



# Enhanced hydrological modelling with the WRF-Hydro lake/reservoir module at Convection-Permitting scale: a case study of the Tana River basin in East Africa

Ling Zhang[1,2], Lu Li[3], Zhongshi Zhang[1,2,3], Joël Arnault[4,5], Stefan Sobolowski[6], Anthony Musili Mwanthi[7,8],

Pratik Kad[3], Mohammed Abdullahi Hassan[8], Tanja Portele[5], Harald Kunstmann[4,5]

[1]Department of Atmospheric Science, School of Environmental Studies, China University of Geosciences, Wuhan 430074, China.
[2]Centre for Severe Weather and Climate and Hydro-geological Hazards, Wuhan 430074, China.
[3]NORCE Norwegian Research Centre, Bjerknes Centre for Climate Research, Bergen 5007, Norway.
[4]University of Augsburg, Institute of Geography, Germany
[5]Karlsruhe Institute of Technology, Institute of Meteorology and Climate Research, Garmisch-Partenkirchen, Germany

[6]Geophysical institute, University of Bergen and the Bjerknes Center for Climate Research, Bergen, Norway

[7]University of Nairobi, Kenya

[8]IGAD Climate Prediction and Applications Center, Nairobi, Kenya

*Correspondence to: Lu Li (luli@norceresearch.no)*

**Abstract.** East Africa frequently faces extreme weather events like droughts and floods, underscoring the need for improved hydrological simulations to enhance prediction and mitigate losses. One of the main challenges in achieving this is low-quality of precipitation data and limitations in modelling skills. Due to drought sensitivity, flood proneness and data availability, the upper and middle stream of the Tana River basin was used as a case to address some of the challenge. We performed convection-permitting (CP) simulations using the Weather Research and Forecasting (WRF) model, and utilizing the CPWRF output as a driver we conducted WRF Hydrological modelling (WRF-Hydro) integrated with the lake/reservoir module. The CPWRF precipitation outperforms the ERA5 using IMERG as the benchmark, particularly for the precipitation amount over mountainous regions and light precipitation events (1-15 mm day$^{-1}$) in the dry seasons. The improved precipitation especially alleviates the peak false, when comparing the well-calibrated lake-integrated model driven by CRWRF output (LakeCal) to that by ERA5, with an NSE increase of 0.53. Additionally, the lake/reservoir module effectively mitigates the model-data bias, especially for dry-season flow and peak flow, when comparing the lake-integrated model (LakeCal) to the model without the lake (LakeNan), with an NSE increase of 1.67. The lake module makes river discharge more sensitive to spin-up time and affects discharge through lake-related parameters. Adjustments to the lake-integrated model's runoff infiltration rate, Manning's roughness coefficient, and the groundwater component have minimal impact on the dry-season flows. Dividing by the total NSE increase, hydrological modelling improvement is 24 % and 76 % from CPWRF simulation and lake module, respectively. Our findings highlight the enhanced hydrological modelling capability with the lake/reservoir module and CPWRF simulations, offering a valuable tool for flood and drought predictability in data-scarce regions such as East Africa.

## 1. Introduction

The credibility of hydrological simulations in data-scarce regions is challenged by low-quality of precipitation data (regarding incomplete and unreliability, and poor in-suit coverage), and limitations of hydrological modelling given the underlay's complexities. To make well-informed decisions with respect to flood/drought adaptation and loss mitigation, elected officials,





planners, and the public require relatively reliable information on flood and drought forecasts, which rely on skilled hydrological
simulations. This issue could be particularly acute in drought/flood-prone and vulnerable areas such as East Africa. The economy
and population in East Africa mainly depend on rain-fed agriculture and pastoralism, which suffers from frequent droughts and
floods (Taye and Dyer, 2024). For example, the drought of 2022 triggered an exceptional food security crisis in Ethiopia, Somalia,
and Kenya, pushing more than 20 million people into extreme hunger (NASA, 2022). Similarly, the flood in 2023 here killed more
than 100 people and displaced over 700,000 (NASA, 2024). The highlighted risk in East Africa urges effective hydrological
simulation for better hydrological extreme forecasts, thus supporting effective water resource planning and management, and
aiding informed decision-making and loss mitigation for officials, planners, and the public.

Obtaining even the present-day precipitation, especially in mountainous regions, is challenging due to poor in-situ coverage, and
incomplete or unreliable records. Such data scarcity even complicates the evaluation of model output (Li et al., 2017). This issue
is only further exacerbated as one decreases grid-spacing to km scales. Gridded precipitation productions tried to be an alternative,
involving merged data [such as Climate Hazards Group InfraRed Precipitation with Station data (CHIRPS) (Funk et al., 2015)],
reanalysis data [i.e. ERA-Interim (Dee et al., 2011)], and satellite-based data [i.e. Tropical Rainfall Measuring Mission (Adjei et
al., 2015) and Integrated Multi-satellite Retrievals for GPM (IMERG) (Dezfuli et al., 2017)]. However, they present uncertainties,
such as false detection of precipitation events and bias of precipitation amount (Bitew and Gebremichael, 2011; Ma et al., 2018;
Dezfuli et al., 2017) limiting their suitability in the hydrometeorological application. The uncertainty is particular in mountainous
regions (Li et al., 2018; Maranan et al., 2020; Zandler et al., 2019). Also, precipitation from coarse-resolution Global Climate
Models shows limitations (Monsieurs et al., 2018; Kad et al. 2023), due to the model configuration, such as resolution and
parameterization, which are crucial for a more realistic representation of processes (Kad et al., 2023a; Tao et al., 2020).

High-resolution dynamical simulation is a promising tool with which one can generate precipitation with realistic regional detail,
due to the capability of capturing realistic regional details, such as topography and local processes that influence orographic effects
(Kad and Ha, 2023; Tao et al., 2020). In Kerandi's research (2017), WRF with a refined resolution of 25 km, better captured annual
and interannual variability and spatial distribution of precipitation in the Tana River basin, than the coarse resolution of 50 km.
Indeed, at relatively coarse resolution (such as >20 km resolution), RCMs generally fail to adequately represent precipitation and
exhibit uncertainties when compared to reanalysis, rain gauges, and satellite observations (Biskop et al., 2012; Ji and Kang, 2013).
A refined horizontal resolution has the potential to significantly improve precipitation simulation over Equatorial East Africa (Pohl
et al. 2011).

Convection-permitting regional climate models (CPRCMs, typically with < 5 km resolution) provide an explicit representation of
convection and thus allow to capture precipitation extremes at the local scale, in comparison to coarse resolution (Kendon et al.,
2021; Schwartz, 2014; Weusthoff et al., 2010). The added value from CPRCMs relative to the parametrized regional climate
models, involves improved representations of the intensity distribution (Senior, 2021; Berthou et al., 2019), diurnal cycle (Stratton
et al., 2018) and storm size and duration (Crook et al., 2019). It is noteworthy that CPRCMs better capture surface heterogeneities
and give more realistic climate simulations over mountains (Kawase et al., 2013; Rasmussen et al., 2014). Additionally, CPRCMs
exhibit increased performance over Africa (Senior, 2021), in presenting rainy events, diurnal cycle and peak time for the Lake
Victoria Basin of East Africa (Lipzi et al. 2023), and sub-daily rainfall intensity distribution (especially those related to the
convective rainfall) in the tropics (Folwell et al. 2022). Therefore, CPRCM could be applied to generate more realistic precipitation
with more regional details in East Africa.




Offline atmosphere-hydrological modelling is a commonly used approach for flood and drought simulation or prediction. Ideally, regional climate model (RCM) output data was directly used in hydrological applications. However, this can cause issues of physical inconsistency (Chen et al., 2011; Teutschbein and Seibert, 2012). A better approach would be to couple atmospheric and hydrological modelling systems to ensure physical consistency. A coupling of the Weather Research and Forecasting Model (WRF) and the WRF hydrological modelling system (WRF-Hydro; Gochis et al., 2018) shows advantages in hydrology simulations and hydrological extremes forecasting globally (e.g., Kerandi et al., 2018; Li et al., 2017), involving urban flood prediction over the Dallas-Fort Worth area of North America (Nearing et al. 024) and drought estimation in South Korea (Alavoine and Grenier 2023). In Africa, WRF-Hydro has also proven useful in discharge simulations in the Ouémé River of West Africa (Quenum et al. 2022) and the Tana River basin (Kerandi et al. 2018). Kerandi's study showed minimal differences in precipitation between the stand-alone and fully coupled, suggesting a limited impact of precipitation recycling and land-atmosphere feedback on soil moisture and discharge in Tana River basin. This could be seen from other regions, such as Crati River Basin in Southern Italy by Senatore et al. (2015) and United Arab Emirates by Wehbe et al. (2019).

Even though WRF-Hydro shows potential, its use over East Africa needs to be refined through the implementation of more comprehensive hydrological processes. Many reservoirs have been built in East Africa (Palmieri et al., 2003), which can change magnitude and timing of natural streamflow, usually attenuating and delaying flows in the rain season, and also releasing water in dry periods (Zajac et al., 2017; Hanasaki et al., 2006). Incorporating lakes/reservoir processes in hydrological simulation is required for a reliable model when applied in the region with lakes (Hanasaki et al., 2006; Lehner et al., 2011). However, only a few hydrological simulations over East Africa are related to lakes (Oludhe et al., 2013; Naabil et al., 2017; Siderius et al., 2018). The study on the impact of reservoirs over East Africa was even fewer, let alone the hydrological modelling with meteorological-hydrological links. Naabil (2017) used WRF-Hydro with the dam-water-balance model for dam-level simulation and water resource assessment in Tono dam basin. However, in this research, the reservoir module was not included in the WRF-Hydro system, preventing accurate capture of dam impact on discharge and other hydrological variables. Therefore, hydrological modelling coupled with its lake/reservoir module is required over East Africa for reliable flood and drought simulations. While the WRF-Hydro system, with its lake/reservoir module, shows promise for simulating water balance affected by reservoirs (Maingi and Marsh, 2002), its use in East Africa, especially in large river basins like the Tana River, remains limited.

The Tana River basin in East Africa is ideal for enhanced hydrological modelling due to its proneness and vulnerability to droughts and floods, as well as the data available. The observational discharge records provide a benchmark for simulations despite some uncertainties. The basin supports vital ecosystem services for Kenya, including drinking water supply, hydro-electric power, agriculture and biodiversity, and is home to eight million people (Lange et al., 2015). However, the region faces increasing risks of drought and flood, which are likely exacerbated by climate change. Droughts occur approximately every five years, causing water shortages for drinking water, irrigation, and fishing (Bonekamp et al., 2018). The flood in 2018, overflowed the bank, damaged crops, homes, and infrastructure, and subsequently displaced thousands of people, contributing to outbreaks of waterborne diseases (such as cholera) (Kiptum et al., 2024). So, robust hydrological modelling in the Tana River basin is essential for accurate predictions of extreme events and risk assessment. Using this basin as a case, the present study aims to address some of the issue related to flood/drought risk mitigations, through a convection-permitting regional climate (CPCRM) simulation using WRF model and a more comprehensive hydrological model using lake-integrated WRF-Hydro system. We target the following sub-objectives: (1) to improve climate output (particularly focusing on precipitation) by CPCRM simulation and using the





enhanced precipitation to advance the hydrological simulation; (2) to explore the potential of lake/reservoir module to improve the hydrological modelling; (3) to build an enhanced WRF-Hydro system and investigate the contributions of the two components to hydrological simulations. The research is to improve hydrological models for better water resource management and risk mitigation, supporting sustainable practices in regions vulnerable to water-related damages.

## 2. Study area and data

Located in the tropics, the Tana River Basin exhibits dual peaks of precipitation over time due to the biannual migration of the Intertropical Convergence Zone (ITCZ). The spatial pattern of the precipitation is profoundly modulated by the basin's varied topography and atmospheric deep convection (Kad et al., 2023; Johnston et al., 2018), resulting in a gradient of arid to semi-humid conditions from the lowlands to the highlands and coastal areas (Knoop et al., 2012). The precipitation is also influenced by El Niño/Southern Oscillation (Otieno and Anyah, 2013; Anyah and Semazzi, 2006), IOD (Williams and Funk, 2011), and rising atmospheric $CO_2$ (Kad et al., 2023).

For data availability, our study focuses on the upper and middle sections of the Tana River Basin (TRB), covering an area of 32,865 km² upstream of Garissa city (S 1.25°~N 0.50°, E 36.50°-E 39.75°). This region includes famous mountain ranges such as the Mount Kenya massif and the Aberdare Range, alongside plain surfaces (Fig. 1 b). The region is characterized by a complex interplay between mountainous terrain and flat surface, with elevation ranging from 34 meters to excess of 4800 (Fig. 1 a). We classified the terrain into mountainous regions above 1,600 meters and plains below 1,600 meters. There are five reservoirs in the basin and along the Tana River (Table 1, Fig. 1 c). It is worth noting that the Garissa station is downstream Rukanga and the lakes between them are Masinga, Kamburu, Gitaru, Kindaruma, and Kiambere from the upstream to downstream. While the lakes don't affect the streamflow at Rukanga, they do impact the discharge at Garissa.



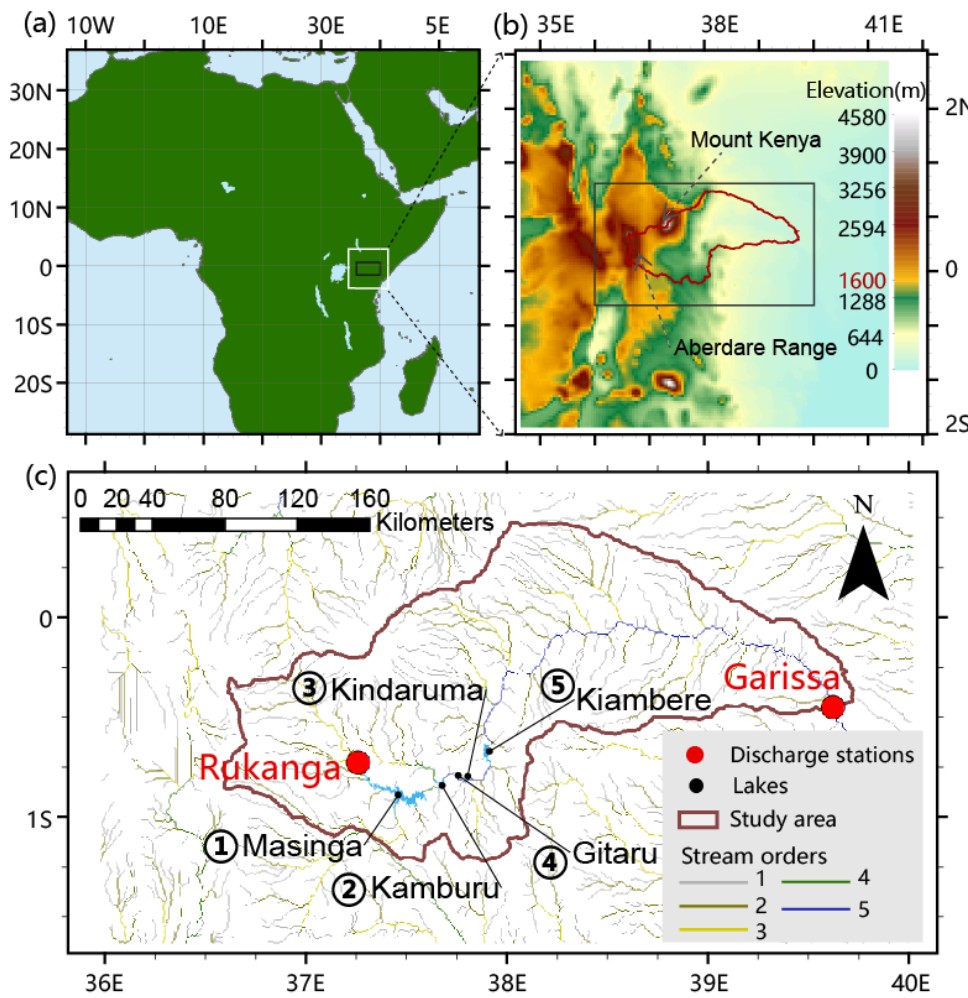

**Figure 1. Study basin location in East Africa. (a) The WRF domain with a resolution of 5 km (shown with the white frame) and the location of the inner region (a black frame) used as the domain of WRF-Hydro simulation (b) A zoomed view of the inner area showing topography, two major mountains, and the basin boundaries. (c) Drainage map of the upper and middle stream of the Tana River Basi, including the discharge stations, lake/reservoir water level stations and the stream orders for hydrological modelling in the WRF-Hydro model system.**

**Table 1. Lakes/Reservoirs in upper and middle Tana River basin (TRB).**

| Name | Water level (max/min; unit: m) | Water depth (m) | Area (km²) | Operating date |
|---|---|---|---|---|
| KAMBURU | 1007/996 | 1007 | 11.7 | 1974 |
| KINDARUMA | 781/775 | 7811 | 2.1 | 1981 |
| MASINGA | 1058/1035 | 1058 | 111.6 | 1981 |
| GITARU | 925/917 | 9255 | 2.7 | 1978 |
| KIAMBERE | 702/681 | 702 | 23.2 | 1981 |

Here, we used a global satellite product of GPM_3IMERGDF (GPM IMERG precipitation version 6 at daily temporal resolution and 0.1° x 0.1° spatial resolution) (Huffman et al., 2020) for WRF precipitation evaluation, downloaded from the NASA website (https://gpm.nasa.gov/data-access/downloads/gpm, accessed on 28 Apr 2023). These climate data cover the period 2010-2014.





Discharge observations during 2011-2014 at two stations in TRB (Garissa and Rukanga), obtained from the Water Resources
Authority of Kenya (WRA), are used for WRF-Hydro model discharge sensitivity analysis and calibration (Fig. 1).
**3.    Methodology**
**3.1.    WRF domain design for convection-permitting WRF modelling**
To obtain convection-permitting modelling precipitation, we used the Advanced Research WRF (WRF-ARW) model of version
4.4 (Skamarock et al., 2019) with the designed domain of 5 km spatial resolution (Fig. 1). The lateral boundaries were forced with
the 6-hourly ERA5 reanalysis with a spatial resolution of 0.25 degrees (Hersbach et al., 2020). The model was set with 50 vertical
levels up to 10hPa and running from 1 January 2010 to 1 January 2015 with the first year of spin-up.

The Grell-Freitas Ensemble Scheme (Grell and Freitas, 2014) was used for the cumulus scheme (which is only for the outer domain,
while the convection parameterization was turned off for the inner domain), the Mellor-Yamada Nakanishi Niino Level 2.5
(MYNN2.5) Scheme (Nakanishi and Niino, 2006) for the planetary boundary layer, the RRTM scheme for longwave radiation
(Mlawer et al., 1997) and the Dudhia Shortwave scheme for shortwave radiation (Jimy Dudhia, 1989). The Noah-MP Land Surface
model ('Noah-MP LSM', Yang et al., 2011) was used for land surface scheme.
**3.2.    Sensitivity analysis and calibration strategy for WRF-Hydro modelling**
**3.2.1.    WRF-Hydro modelling system and preliminary calibration**
For hydrological modelling, WRF-Hydro system (Gochis et al., 2018) of Version 5.3. was employed in an offline mode, using the
CPWRF atmospheric simulations within a domain at 5 km resolution with 90×50 pixels over the TRB as the driver (Fig. 1). The
sub-grid routing processes were executed at a 500 m grid spacing and surface physiographic files were generated by ArcGIS 10.6
(Sampson and Gochis, 2015). The physiographic files included high-resolution terrain grids that specified the topography, channel
grids, flow direction, stream order (for channel routing), a groundwater basin mask and the position of stream gauging stations
(Fig. 1c). the first five stream orders are shown in Fig. 1c. We activated the saturated subsurface overflow routing, surface overland
flow routing, channel routing and base-flow modules. The overland flow routing and channel routing were calculated by a 2-D
diffusive wave formulation (Julien et al., 1995) and a 1-D variable time-stepping diffusive wave formulation, respectively.

The model involves the five lake/reservoirs using a level-pool lake/reservoir module which calculates both orifice and weir outflow.
Fluxes into a lake/reservoir object occur when the channel network intersects a lake/reservoir object. The level-pool scheme tracks
water elevation over time, and water out of the lake/reservoir exits either through weir overflow
($Outflow_w$) or orifice-controlled flow ($Outflow_o$) following Eq. (1) and (2).
$$Outflow_w = \begin{cases} C_w L h^{3/2}, & h > h_{max} \\ 0, & h \leq h_{max} \end{cases}$$     (1)
where $h$ is the water elevation (m), $h_{max}$ is the maximum height before the weir begins to spill (m), $C_w$ is the weir coefficient, and
$L$ is the length of the weir (m).
$$Outflow_o = C_o S_o \sqrt{2gh}$$     (2)
where $C_o$ is the orifice coefficient, $S_o$ is the orifice area ($m^2$), and $g$ is the acceleration of gravity (m s$^{-2}$).






For the sensitivity analysis and model optimization, we initially calibrated the WRF-Hydro system without the lake/reservoir
module. Two key hydrological parameters, REFKDT and MannN, were tuned using the auto-calibration Parameter Estimation
Tool (PEST, http://www.pesthomepage.org). The optimization is performed by maximizing the discharge simulation accuracy,
indicated by Nash-Sutcliffe Efficiency (NSE) coefficient (Nash and Sutcliffe, 1970) of the Garissa discharge. The primarily
calibrated model was mentioned as LakeNan in the following.
**3.2.2. Experiments designed for sensitivity analysis in WRF-Hydro system modelling with lake/reservoir module**
To optimize WRF-Hydro modelling over TRB, we facilitated a comprehensive sensitivity analysis, involving spin-up time,
hydrological parameters, groundwater components, and lake-related parameters. Groundwater component tunning focuses on the
parameter GWBASEWCTRT (an option for groundwater mode). Hydrological parameters include Manning roughness parameter
(MannN) and runoff infiltration coefficients (REFKDT). Lake-related parameters cover the elevation of maximum lake/reservoir
height (LkMxE, unit: m), weir elevation (WeirE, unit: m), weir coefficient (WeirC, ranging from zero to one), weir length
(WeirL, unit: m), orifice area (OrificeA, unit: $m^2$), orifice coefficient (OrificeC, ranging from zero to one), orifice elevation
(OrificeE, unit: m), and lake/reservoir module area (LkArea, unit: $m^2$).

For sensitivity analysis of the specific parameter, we conducted a set of experiments. In each experiment, only the focused
parameter was changed while others were maintained at their default (Table 2). The defaults of lake-related parameters were
obtained from WRF-Hydro GIS pre-processing toolkit (Gochis et al., 2018), while the others were obtained from the preliminary
calibrated WRF-Hydro without lake/reservoir module (LakeNan, Sect. 3.2.1).
**Table 2. The default values for sensitivity experiments.**

| Group | Parameters | The default value |
|---|---|---|
| Others | Spin-up time | restart with a 10-year spin-up time using the initial file from a 10-year simulation covering January 2005 to December 2014. |
| Hydrological parameters | REFKDT | 5 |
| | MannN | (0.55,0.35,0.15,0.1,0.07, 0.05, 0.04, 0.03, 0.02, 0.01) for the ten stream orders |
| Groundwater | GWBASEWCTRT | "GWBASESWCRT_Sink" for sensitivity tests of spin-up and hydrological parameters; "GWBASESWCRT_Passthrough" for sensitivity tests of lake-related parameters, and the subsequent calibration. |
| Lake-related parameters | LkMxE | -9,957,781,074,917,690 |
| | WeirE | (990.5,775.9,1067.9,915.3,689.1) |
| | WeirC | (0.4,0.4,0.4,0.4,0.4) |
| | WeirL | (10,10,10,10,10) |
| | OrificeA | (1,1,1,1,1) |
| | OrificeC | (0.1,0.1,0.1,0.1,0.1) |
| | OrificeE | (965,764,1033.3,905.7,644.3) |
| | LkArea | (11.7,2.1,111.6,2.7,23.2) |

REFKDT and MannN default values are from the preliminary calibration for LakeNan model. The MannN value is different for each stream
order from 1 to 10. (Value1, Value2, Value3, Value4, Value5) indicate value for the five reservoirs (KAMBURU, KINDARUMA, MASINGA,
GITARU, KIAMBERE), obtained from WRF-Hydro GIS pre-processing toolkit. Two options for the groundwater component were involved in
the experiments. Groundwater component with "GWBASESWCRT_Sink" option creates a sink at the bottom of the soil column and water
draining from the bottom of the soil column leaves the system into the sink, while that with "GWBASESWCRT_Passthrough" bypasses the
bucket model and dumps all flow from the bottom of the soil column directly into the channel.
**Sensitivity to spin-up time**





To obtain a stable hydrological simulation, a spin-up time is required. Insufficient spin-up for initialization introduces unnecessary
uncertainty into hydrological simulations, which may affect the subsequent sensitivity analysis and hydrological modelling
assessments. Previous studies have shown that spin-up time affects initial conditions such as the soil moisture content, surface
water, lake/reservoir module water level, and groundwater, which subsequently influences the fidelity of model simulations (Ajami
et al., 2014a; Ajami et al., 2014b; Bonekamp et al., 2018; Seck et al., 2015). For example, groundwater simulation even needs more
than 10 years-spin-up to get stable (Ajami et al., 2014a). Since the shortest spin-up time likely depends on the quality of the model
input (especially soil data) and likely on local conditions, the impact of the spin-up time needs to be assessed on per-case basis.
Therefore, we first investigated the spin-up time sensitivity to get the shortest time for stable modelling and computable saving.
In our study, we conduct experiments of 17 different spin-up times (Table 3) to investigate their impacts on peak flow, average
discharge, and water levels of reservoirs in TRB, respectively for WRF-Hydro systems with (LakeRaw) and without lake/reservoir
module (LakeNan). To analyze the sensitivity of peak flow, we designated the starting point of the simulation as the observed
peak-flow day (26 November 2011), with spin-up times ranging from 1 day to 12 years. In the spin-up experiments, the restart date
precedes January 1th 2010 which is absent in WRF drivers, so we employ data in 2010 substituting the driving climate for each
preceding year (i.e. 2000, 2001,…,2009). In all experiments of LakeRaw, the parameters are set as the default shown in Table 2.
**Table 3. Overview of 17 spin-up time experiments**

| Experiment name | Restart date | Spin-up time |
| --- | --- | --- |
| 1 spin-up | 25 November 2011 | 1 day |
| 3 mon spin-up | 26 November 2011 | 3 months |
| 6 mon spin-up | 26 May 2011 | 6 months |
| 9 mon spin-up | 26 February 2011 | 9 months |
| 1 year spin-up | 26 December 2010 | 1 year |
| 15 mon spin-up | 26 August 2010 | 15 months |
| 18 mon spin-up | 26 May 2010 | 18 months |
| 21 mon spin-up | 26 February 2010 | 21 months |
| 3 year spin-up | 1 January 2009 | 3 years |
| 4 year spin-up | 1 January 2008 | 4 years |
| 5 year spin-up | 1 January 2007 | 5 years |
| 6 year spin-up | 1 January 2006 | 6 years |
| 7 year spin-up | 1 January 2005 | 7 years |
| 8 year spin-up | 1 January 2004 | 8 years |
| 9 year spin-up | 1 January 2003 | 9 years |
| 10 year spin-up | 1 January 2002 | 10 years |
| 11 year spin-up | 1 January 2001 | 11 years |
| 12 year spin-up | 1 January 2000 | 12 years |

The initialization time for one model to reach equilibrium was calculated as the time required for the temporal changes in the
model output variable to decrease to a specific threshold value (Cosgrove et al., 2003). In our study, this threshold value was set
as half the standard deviation of hydrological variables from the last experiments (i.e. 9, 10, 11, and 12-year spin-up experiments).
The temporal changes were measured as the difference of a hydrological variable between the two adjacent experiments.
**Sensitivity to hydrological parameters**
MannN and REFKDT have been demonstrated to significantly influence the simulated river discharge (Ryu et al., 2017; Yucel et
al., 2015). Therefore, REFKDT and MannN for the first five stream orders were chosen for the sensitivity test, separately. For each



of the tests, the parameter values range from minimum to maximum, creating ten values with nearly equal intervals and generating
ten experiments (Table 4). Among them, MannN should be larger than 0, so the minimum scaling was 0.1, instead of 0.
**Table 4. Sensitivity analysis (SA) experiments** designed for the two key hydrological parameters REFKDT.

| Experiments for REFKDT SA | Value |
|---|---|
| REFKDT_1 | 0.02*default |
| REFKDT_2 | 0.13*default |
| REFKDT_3 | 0.24*default |
| REFKDT_4 | 0.35*default |
| REFKDT_5 | 0.46*default |
| REFKDT_6 | 0.56*default |
| REFKDT_7 | 0.67*default |
| REFKDT_8 | 0.78*default |
| REFKDT_9 | 0.89*default |
| REFKDT_10 | 1*default |

Note: The default is obtained from the WRF-Hydro GIS pre-processing toolkit. * indicates multiplication.
**Table 5. Sensitivity analysis (SA) experiments** designed for the two key hydrological parameters MannN of the first five stream orders.

| Experiments for MannN SA | Value |
|---|---|
| MannN_1 | 0.1*default |
| MannN_2 | 0.44*default |
| MannN_3 | 0.89*default |
| MannN_4 | 1.33*default |
| MannN_5 | 1.78*default |
| MannN_6 | 2.22*default |
| MannN_7 | 2.67*default |
| MannN_8 | 3.11*default |
| MannN_9 | 3.56*default |
| MannN_10 | 4.00*default |

Note: The default is obtained from the WRF-Hydro GIS pre-processing toolkit. * indicates multiplication.
**Sensitivity to groundwater component**
We investigate the sensitivity of groundwater components by tunning GWBASWCTRT, with two options in two experiments.
Groundwater component with "GWBASESWCRT_Sink" option creates a sink at the bottom of the soil column and water draining
from the bottom of the soil column leaves the system into the sink, while that with "GWBASESWCRT_Passthrough" bypasses
the bucket model and dumps all flow from the bottom of the soil column directly into the channel. It's important to note that with
the option "GWBASESWCRT_Sink", water draining from the bottom of the soil column will not achieve water balance closure.
**Sensitivity test of lake/reservoir parameters**
Morris (Morris, 1991) was employed to analyze the sensitivity order of the seven lake-related parameters, due to its low
computational cost and ease of interpretation (Wei, 2013), which is widely used as a global sensitivity analysis method in
hydrological models, particularly in computationally expensive models (Song et al., 2013; Wei, 2013). In the study, the sensitivity
analysis was simultaneously performed on the five lakes to reduce computational cost. In the Morris experiment, the eight main
lake-related parameters of the five lakes were normalized to a range of 0-1, by subtracting the minimum value and dividing by the
maximum minus the minimum (Table 5). Based on the eight normalized values with a lower value of zero and an upper of one,
we generated all samples for Morris screening, where the number of replications R, level p and sample size N were set as 10 and


252 4, and 90 (i.e. 90 parameter sets for 90 runs), respectively. For each sample (corresponding to a WRF-Hydro simulation), the eight

253 parameters for each lake/reservoir were obtained by inverse-normalization. The other parameters were kept as default. Parameter

254 sensitivity was evaluated by analyzing the influence of parameter change on varying degrees of model output, which was measured

255 by the order of importance (Francos et al., 2003).

256 **Table 6. Sensitivity analysis experiments designed for the 8 lake/reservoir-related parameters.**

| Parameters | Value_min | Value_max |
|---|---|---|
| OrificeC | 0.01*default | 10*default |
| WeirL | 0.01*default | 1.2*default |
| WeirC | 0.001*default | 0.25*default |
| OrificeA | 0.001*default | 1000*default |
| Dam_Length | 0.001*default | 20*default |
| LxMkE | Wlmax-Wd*0.5 | Wlmax+Wd*0.5 |
| WeirE | OrificeE_default | Wlmax+Wd*0.5 |
| OrificeE | Wlmin*0.5 | Wlmin |

257 Note: Wlmax, Wlmin, Wd, and OrificeE_default indicate the max water level, min water level, water depth, and OrificeE default value,
258 respectively. The default is obtained from WRF-Hydro GIS pre-processing toolkit.

259 We also compared the sensitivity among the five lakes to simulated discharge. To conserve computing resources, the test was

260 conducted based on the simulations from the calibration. For each lake test, there is a set of more than 30 simulations. Each of the

261 sets involves the seven parameters (LkMxE, WeirE, OrificeE, WeirC, WeirL, OrificeC, and Damlength). In all the sets, the values

262 of seven parameters synchronously change linearly from the minimum to the maximum shown in Table 6.

263

264 In the parameter setting, we make some rules to constrain the three parameters (LkMxE, WeirE, and OrificeE), to make the

265 simulation result reasonable: (1) LkMxE should be larger than WeirE and OrificeE; (2) OrificeE was suggested to be smaller than

266 WeirE. To satisfy these constraints, the OrificeE is set to be below the minimum water level, WeirE ranges from the OrificeE

267 default value to the maximum water level plus half water depth, and LkMxE changes from the maximum water level minus haft

268 depth to maximum water level plus half depth (Table 1). Besides, OrificeC and WeirC should be kept between zero and 1 which

269 should be a constraint. The setting of maximum and minimum values, and experiment count are flexible, provided they make sense

270 and the simulation result is reasonable.

### 271 3.2.3. Final calibration for WRF-Hydro system modelling with lake/reservoir module

272 Based on the sensitivity analysis, we developed a comprehensive calibration strategy for the WRF-Hydro system incorporating the

273 lake/reservoir module. Based on the preliminary calibration (Sect. 3.2.1), we re-tuned the lake-related parameter sets for the five

274 lakes. Each lake was calibrated sequentially from upstream to downstream, with its parameter set undergoing more than 30

275 experimental iterations. Once the upstream lake/reservoir was calibrated, its parameters were fixed as the optimized, and we

276 proceeded to calibrate the parameters for the next downstream lake. Subsequently, we focused on re-tuning REFKDT and MannN,

277 each subjected to 30 experimental iterations. The parameter sets for each experimental iteration were generated according to Sect.

278 3.2.2. Throughout the step, we get a well-calibrated WRF-Hyro model (LakeCal) with the optimal parameter set of the best NSE,

279 calculated over Garrissa discharge from January 2011 to December 2014 against the observation.

### 280 3.3. Peak flow, dry-season flow and rain-season flow

281 To measure modelling performance, we obtained the flow from the long rain season of March-May (MAM) and short rain season

282 of October-December (OND), and the dry season of January-February (JF) and June-September (JJAS), as well as the peak flow.



The max of the daily discharge over 2010-2014 at Garissa station occurred on 26 November 2011 (844 m$^3$ s$^{-1}$) and is used as a
peak-flow case for evaluation. Since the model cannot capture the peak at the exact date, the simulated peak flow corresponding
to the observation, s set as the largest daily discharge during the 21 days which covers the observed peak in the center. The peak
in a certain year is set as the largest daily discharge during this year. Additionally, water level observations from five lakes within
the TRB, obtained from Kenya's Ministry of Energy (KenGen) for the period 2011-2014, are used to assist in model sensitivity
analysis and calibration.
**4.    Results**
**4.1.    WRF Precipitation refinement**
Using IMERG precipitation as a benchmark, we assessed the performance of convection-permitting WRF precipitation at a 5 km
resolution in TRB, through the comparison to ERA5 reanalysis (the input of our WRF simulation). The evaluation focused on
average seasonal precipitation during the long rain season (MAM) and short rains (OND) from 2010-2014. Here, we also calculated
precipitation bias for WRF and ERA5 against IMERG as shown in Fig. 2.

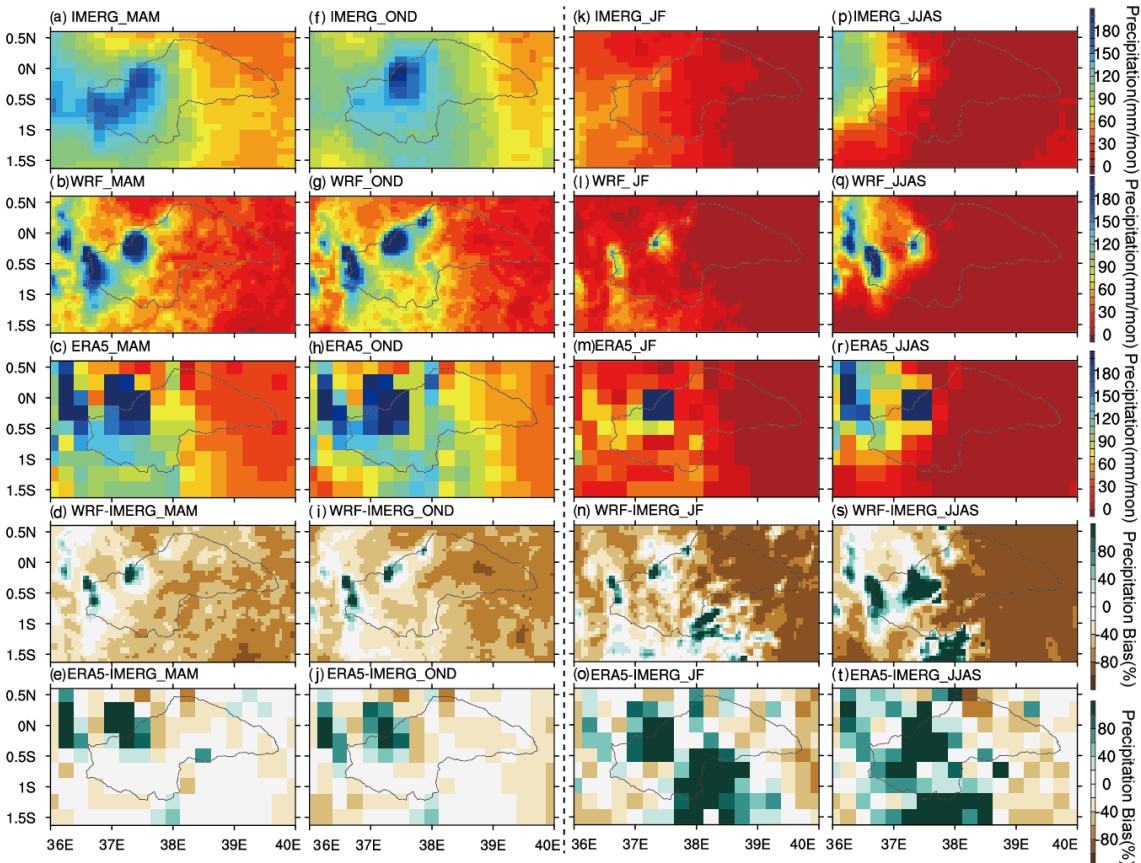


**Figure 2. Season precipitation of March-May (MAM, long rain season, a-c), October-December (OND, short rain season, f-h), and the JF (January-February, k-m), and JJAS (June-August, p-r) over the upper and middle stream of Tana River Basin (TRB), as well as its bias (d-e, i-j, n-o, and s-t). (a, f, k, p), (b, g, i, q), and (c, h, m, r) indicate IMERG, WRF, and ERA5 data. (d, i, n, s) and (e, j, o, t) donates the bias of WRF and ERA5 against IMERG. The seasonal precipitation (MAM, OND, JF, and JJAS) is calculated based on daily data (in March-May, October-December, January-February, and June-August) over 2010-2014. The gray polygon indicates the boundary of the upper and middle sections of the Tana River basin.**

The WRF model captures the spatial pattern of precipitation and its seasonal variations over TRB presented in IMERG (Fig. 2 and
Table 7). WRF simulation shows that the precipitation is primarily concentrated in mountainous regions (such as Mount Kenya
and Aberdare Range in Fig. 1 a), with significantly less precipitation in the plain area (Fig. 2). The annual mean precipitation is
approximately 1500 mm in the mountainous terrain compared to less than 500 mm in the plain area (Table 7). During the rain
seasons (MAM and OND), the total precipitation is 976 mm a$^{-1}$ over the terrain area and 327 mm a$^{-1}$ over the plain area, in contrast
with 417 mm a$^{-1}$ and 33 mm a$^{-1}$ during the dry season (JF and JJAS). This spatial and seasonal pattern is also reflected in IMERG
data (Figs 2 a, f, k, and p), indicating a distinct orographic and seasonal dominance. WRF-simulated precipitation exhibits a smaller
model-data bias in the mountainous areas compared to the plains and during the wet period compared to the dry seasons. The bias
in precipitation over the mountainous area is 47 % (133 mm a$^{-1}$) in dry seasons and 8 % (77 mm a$^{-1}$) in wet seasons, while in the
plains, it is -49 % (-33 mm a$^{-1}$) and -46 % (-279 mm a$^{-1}$). The better skill over the mountain area is more pronounced during the
wet season, with a bias of 4% compared to -45 % in the dry season. Compared to ERA5, WRF precipitation shows better
performance over mountainous areas. For example, the model-data bias from WRF is 210 mm a$^{-1}$ (18 %) for the whole year, while
ERA5 shows a bias of 681 mm a$^{-1}$ (58 %) as shown in Table 7. During the rain season of MAN or OND, WRF's bias is 29 mm a$^{-}$





$^{1}$ (7 %) or 48 mm a$^{-1}$ (10 %), whereas ERA5's is 161 mm a$^{-1}$ (37 %) or 100 mm a$^{-1}$ (22 %). Moreover, the area with the larger bias
(with bias exceeding 60 %) from WRF simulation is much smaller than ERA5. In MAM, OND JF, and JJAS, the regions with
larger biases are 618.2 km$^2$ (1.9 %), 711.0 km$^2$ (2.2 %), 680.0 km$^2$ (2.1 %), and 3431.0 km$^2$ (10.4 %) respectively, while ERA5's
corresponding areas are 1545.5 km$^2$ (4.7 %), 1545.5 km$^2$ (4.7 %), 10818.3 km$^2$ (32.9 %), and 8500.1 km$^2$ (25.9 %). Although a
slightly larger negative precipitation bias exists in the plain area, WRF precipitation doesn't show significantly decreased kills
compared to ERA5 (Table 7).
**Table 7. Seasonal and annual precipitation averaged over the terrain (elevation > 1600 mm) and plain (elevation < 1600 mm) area.**

| Precipitation (mm) | terrain Area | | | | | Plain Area | | | | |
|---|---|---|---|---|---|---|---|---|---|---|
| | Annual | MAM | OND | JF | JJAS | Annual | MAM | OND | JF | JJAS |
| WRF | 1393 | 505 | 471 | 87 | 330 | 359 | 153 | 174 | 16 | 17 |
| ERA5 | 1864 | 557 | 603 | 230 | 474 | 593 | 219 | 278 | 48 | 49 |
| IMERG | 1183 | 457 | 442 | 91 | 193 | 669 | 279 | 326 | 36 | 28 |
| WRF-IMERG | 210(18%) | 48(10%) | 29(7%) | -5(-5%) | 138(72%) | -310(-46%) | -126(-45%) | -152(-47%) | -20(-56%) | -11(-39%) |
| ERA5-IMERG | 681(58%) | 100(22%) | 161(37%) | 139(152%) | 281(146%) | -75(-11%) | -61(-22%) | -48(-15%) | 12(34%) | 22(79%) |

Note: Precipitation from IMERG is the benchmark to evaluate that from WRF simulation.
Monthly averaged precipitation from WRF simulation, calculated over 2010-2014, aligns well with IMERG data (Fig. S1). The
precipitation from WRF well captures the wet-dry season pattern, with precipitation largely falling during long (MAM, 219 mm a$^{-}$
$^{1}$, 40 % of the total annual precipitation) and short rains (OND, 229 mm a$^{-1}$, 42 %) over the TRB. WRF accurately shows the
rainfall peaks in April during the long rain season and November during the short rain season, with simulated values of 95 mm and
178 mm per month, respectively. While both WRF and ERA5 display positive biases in rain seasons and negative biases in dry
seasons against IMERG, WRF offers improved precipitation estimates, distinct in mountainous areas. In the mountainous region,
the WRF-simulated results exhibit superior agreement against IMERG, compared to ERA5 (Figs. 2 d-e, i-j, n-o, s-t, and S1). The
determined coefficient (r$^2$) and biases of WRF-simulated monthly precipitation against IMERG, are 0.71 and 18 mm per month
(15 % of IMERG's regional average), compared to 0.21 and 57 mm per month (58 %) for ERA5 (Table S1). The decreased WRF-
IMERG bias indicates that WRF simulation could alleviate the overestimation from ERA5 in the mountain area, and thus refine
precipitation. Despite no significant improvement in the plain area, no apparent decreased skill exists in WRF simulation, compared
to ERA5.

The probability distribution of regionally averaged daily precipitation from WRF simulation during 2010-2014 (Fig. 3 and Table
8) also exhibits reasonable correspondence to IMERG. Both WRF and ERA5 overestimate the small precipitation events (0-20
mm day$^{-1}$) and underestimate extreme precipitation events (> 20 mm day$^{-1}$), against IMERG. However, WRF aligns more closely
with IMERG for the small precipitation events and extreme precipitation events, particularly for the light precipitation events (1-
15 mm day$^{-1}$) during dry seasons (Fig. 3 c-d and Table 8). The probability of 1-15 mm day$^{-1}$ events from WRF is 0.24, compared
to ERA's 0.49 and IMERG's 0.26. This improvement is also observed in the rain seasons, although not as pronounced. The WRF
simulated probability of light precipitation events is 0.34, compared to 0.66 and 0.38 from ERA and IMERG, respectively.





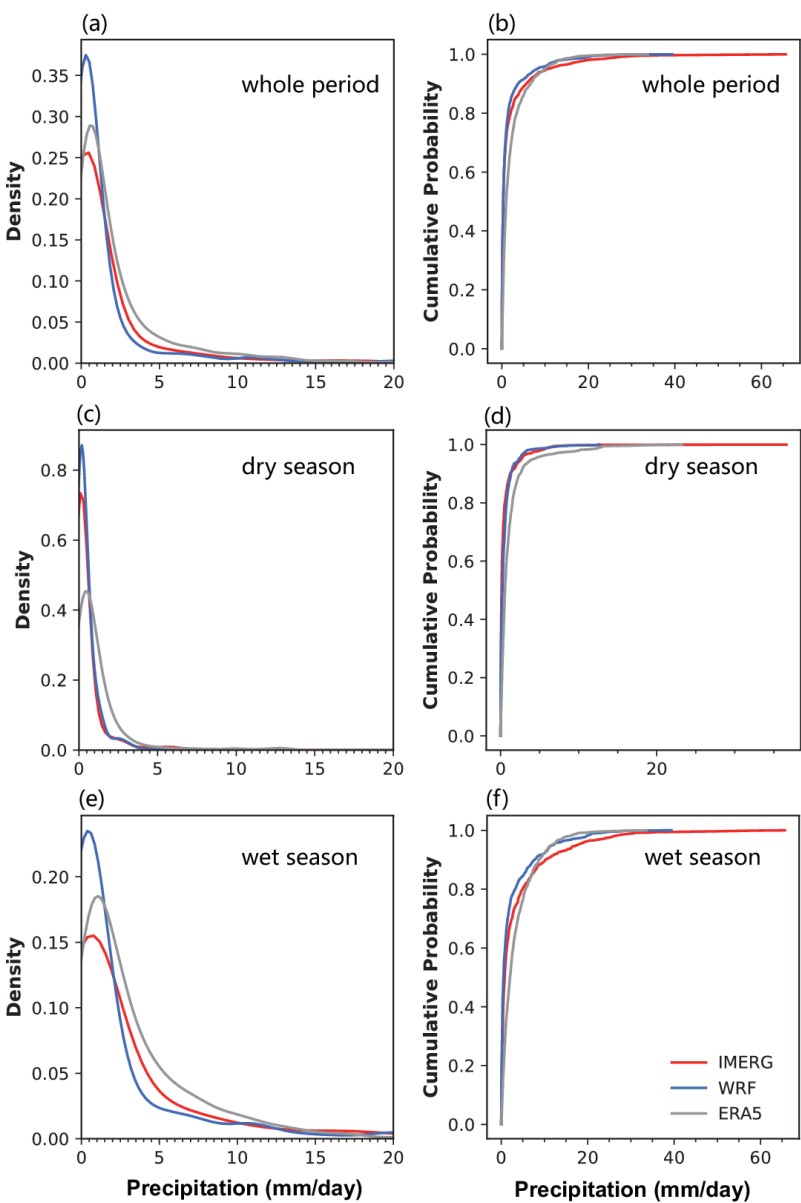


**Figure 3. The distribution (a, c and e) and cumulative distribution (b, d and f) of daily precipitation from WRF-simulation, ERA5, against the IMERG (2010-2014) over the whole period, dry season and wet season. (a, b), (c, d) and (e, f) indicate the daily precipitation distribution over the whole period, dry season and wet season, respectively.**

**Table 8. Cumulative distribution of daily precipitation regionally averaged over TRB, from WRF simulation, IMERG, and ERA5.**

| Precipitation (mm day⁻¹) | Whole period | | | Dry period | | | Wet period | | |
|---|---|---|---|---|---|---|---|---|---|
| | IMERG | WRF | ERA5 | IMERG | WRF | ERA5 | IMERG | WRF | ERA5 |
| 0–20 | 0.981 | 0.991 | 0.995 | 0.999 | 0.999 | 0.999 | 0.962 | 0.982 | 0.991 |
| >20 | 0.019 | 0.009 | 0.005 | 0.001 | 0.001 | 0.001 | 0.038 | 0.018 | 0.009 |
| 1–15 | 0.255 | 0.242 | 0.489 | 0.126 | 0.146 | 0.317 | 0.381 | 0.337 | 0.658 |






Despite some deviation of the daily fluctuations between WRF simulation and IMERG, it is important to recognize that the IMERG
itself has its uncertainties in representing precipitation over East Africa. These include low-intensity false alarms and
overestimating rainfall amount from weak convective events (Maranan et al., 2020). Therefore, we believe that the potential
advantages of the WRF simulation are likely greater than what we have demonstrated by our result. However, using IMERG as
the benchmark, the WRF simulation exhibited a significant improvement, with the model-data bias of 15 % over mountainous
areas compared to 58 % from ERA5, despite slightly degraded performance over the plains (Table S1). Future work could benefit
from incorporating more reliable observational data to enhance precipitation evaluation.

### 4.2.  WRF-Hydro model optimization with lake module

### 4.2.1.  A preliminary investigation of the lake/reservoir impact on discharge

To assess the impact of the lake/reservoir module on hydrological simulation, we compared simulated discharges from different
WRF-Hydro modelling experiments against the observations. These included WRF-Hydro with (LakeRaw) and without the
lake/reservoir module (LakeNan) shown in Fig. 4 and Table S2. The WRF-Hydro model with lake/reservoir module (LakeRaw)
improves discharge simulation compared with the version without (LakeNan), even without model calibration. LakeRaw achieved
an NSE of 0.01 and a bias of 40 %, compared to -1.09 and -53 % from the LakeNan. The inclusion of lake/reservoir module
addresses the underestimation of dry-season flow. However, the lake/reservoir module (in the LakeRaw) tends to induce
overestimation, particularly during the dry season of February-March and August-September which amounts to approximately 81 %
of the annual average dry-season flow. The overestimation in LakeRaw is likely due to uncalibrated parameters, including spin-up
time, the hydrological parameters, groundwater component, and lake-related parameters. The hydrological parameters, which were
based on the model without lake/reservoir module (LakeNan), and the groundwater component and lake-related parameter set as
the default from GIS pre-41.processing (Methodology), need to be re-tuned when the lake/reservoir is included in WRF-Hydro
system. To further improve the WRF-Hydro modelling with lake/reservoir module, the potential of the above parameters was
explored.



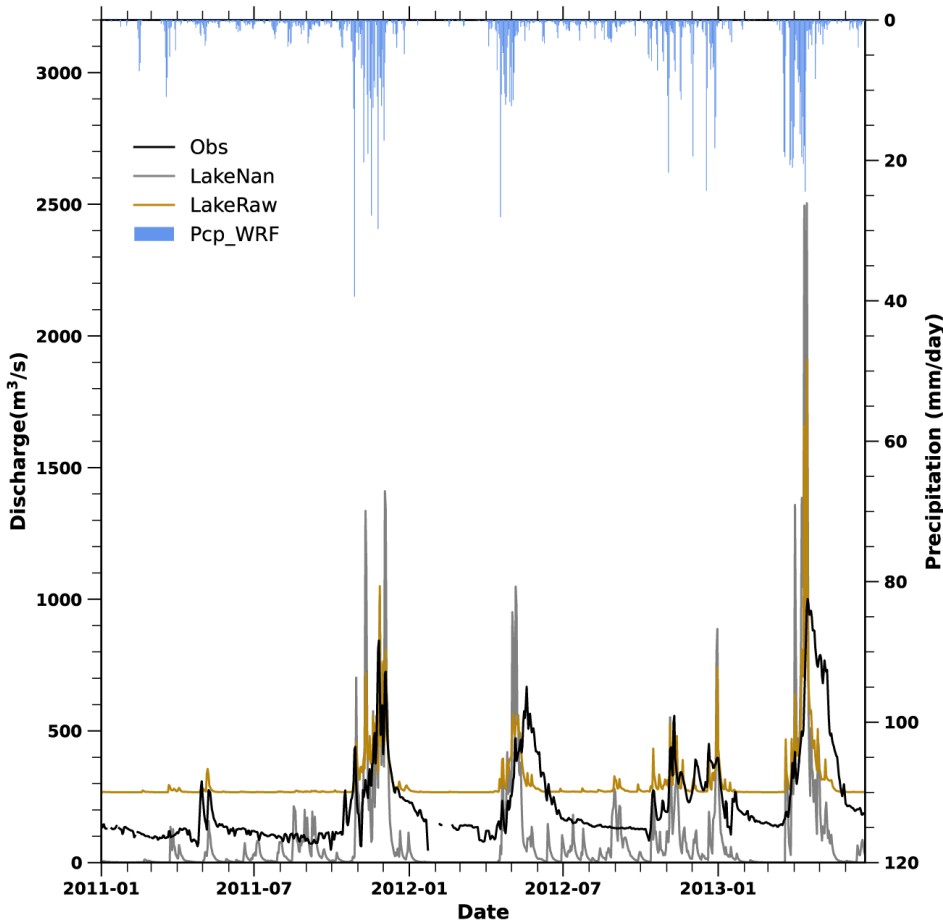

**Figure 4. The simulated daily discharges from WRF-Hydro modelling without the lake/reservoir module (LakeNan, the grey line) and that with the lake/reservoir module using parameters from the LakeNan (LakeRaw, the brown line) against the observations (the black line), as well as the daily precipitation from convection-permitting WRF simulation (Pcp_WRF, the blue bar).**

**4.2.2.    Spin-up time**

The spin-up sensitivity is highlighted in the evolution of discharge during 2011-2014 from the 17 spin-up experiments (Fig. 5 and Table 3). The simulated discharge at the Garissa station on the first day (26 November 2011, the observed peak-flow day) differs between almost every experiment. More specifically, the simulated peak-flow at the Garissa station decreases as the spin-up time gets shorter, which reaches 485 $m^3\ s^{-1}$ in the 12-year spin-up experiment (12y spin-up in Fig. 5a) but only 211 $m^3\ s^{-1}$ in the 1-day spin-up experiment (1d spin-up) from the LakeRaw simulation. The reduction of first-day discharge suggested that, without enough spin-up time, runoff is compensated more to soil moisture and groundwater which hasn't yet reached equilibrium. Generally, runoff of the simulated peak-flow becomes slightly larger with increased spin-up time, until the 6-year spin-up (Fig. 5 b). The simulated average discharge also shows distinct sensitivity to different spin-up times (Figs. 5 d-e). The average discharge at Garissa over the whole, wet and dry seasons during 2011-2014 increased from the underestimation of -49 %, -44 % and -52 % in 1-day spin-up experiment to the overestimation of 21 %, 54 % and 7 % in 12-year spin-up experiment, respectively. It generally takes approximately four years of initialization for the annual discharge at the Garissa station to stabilize. (Figs. 5 d-e).



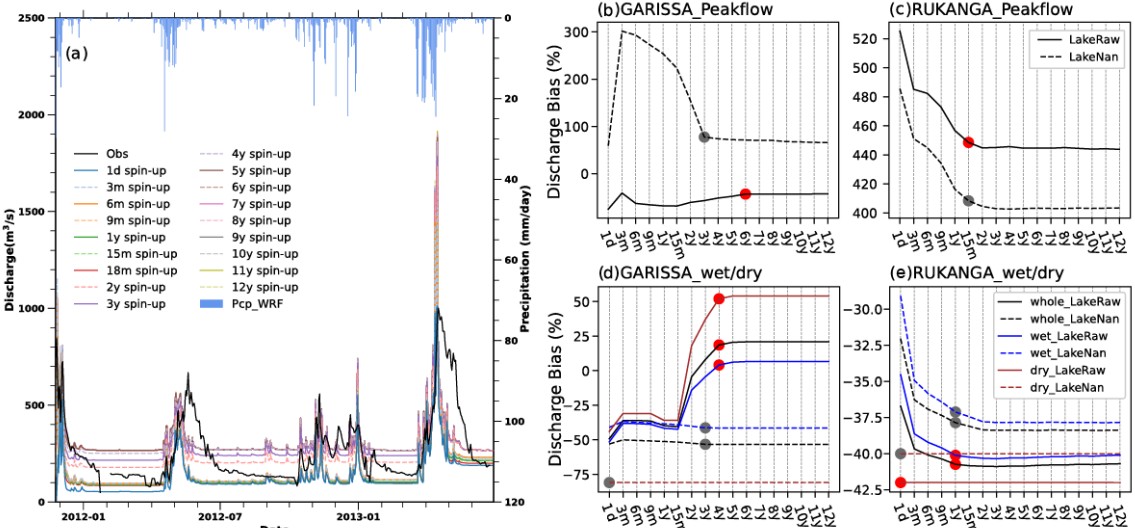

**Figure 5. Sensitivity analysis results from 17 different spin-up experiments. (a) indicates the simulated discharge with spin-ups (the colored lines) ranging from 1 day (1d spin-up) to 12 years (12y spin-up), against the observations (Obs, the black line). The blue bars indicate the daily precipitation from convection-permitting WRF simulation. (b-e) donates the model-data bias of simulated discharge at Garissa (a and c), Rukanga (b and d) with the increase of spin-up time, which are from LakeNan (WRF-Hydro simulation with lake/reservoir module, solid line) and LakeRaw (WRF-Hydro simulation without lake/reservoir module using parameters from LakeNan, dashed line) for the whole year (black line), wet season (MAM and OND, blue line) and dry season (JF and JJAS, red line). The dots indicate the spin-up time required for LakeRaw (red) or LakeNan (grey) to reach equilibrium. Therein, peak-flow (Peakflow) is the largest daily discharge during the 21 days which covers the observed peak (largest observed daily discharge over 2011-2014) in the center.**

The initial time differs spatially, with shorter spin-up in the upstream area than in the downstream. In the LakeNan simulation, the initialization time of discharge metrics (i.e. peak-flow, average discharge, rain-season flow, and dry-season flow) at Rukanga station upstream is less than 2 years but could be 3 years at Garissa station downstream. The longer spin-up in the downstream area might be ascribed to the larger drainage area which needs a longer convergence time, compared to the upstream. The prolongation of spin-up time is more distinct in the simulation with lake/reservoir module than the one without. In the LakeRaw simulation, the initialization time at the upstream (Rukanga station) remains less than 2 years for discharge metrics, while the initialization time for peak-flow at the downstream (Garissa station) extends to 6 years. This stronger prolongation of spin-up time indicates the lake/reservoir affection.

Lake/reservoir module seems to prolong the necessary spin-up time for the downstream area (Fig. 5b). Besides the peak-flow, the spin-up time for whole-period, dry-season, and rain-season flow is prolonged to 4 years in the LakeRaw simulation, compared to the 3, 0 and 3 years in the LakeNan simulation. The larger spin-up difference in dry-season discharge between the LakeRaw (3 years) and LakeNan (0 years) simulations demonstrate a larger sensitivity of dry-season to the lake/reservoir module, compared to the rain-season.

The water levels of the five lakes show the same spin-up time. However, a larger lake seems to require more time to reach equilibrium. The lakes are interconnected, so the initialization time is determined by the longest spin-up. Therefore, despite the disparate sizes, the initialization times of the five lakes are the same. The bias from the LakeRaw simulation is considerable (> 80 %). This is due to that the parameters used in the LakeRaw are from the primarily calibrated LakeNan Model or GIS pre-processing (Methodology), which needs further calibration for the WRF-Hydro system.



#### 4.2.3. Sensitivity analysis from hydrological parameters


The MannN parameter exhibits a substantial impact on the peak flow, with lower values corresponding to higher discharge peaks
(Fig. 6 a and Table S3). As MannN scale decreases from 4 to 0.1, the average discharge at Garissa increases from 294 m$^3$ s$^{-1}$ to
297 m$^3$ s$^{-1}$ and peak-flow increases from 975 m$^3$ s$^{-1}$ to 1309 m$^3$ s$^{-1}$. In addition, the smaller MannN value delays the arrival of peak
flows, shifting the peak-flow date from 6 December 2011 to 2 December, advancing by four days, with MannN ranging from being
scaled up by 4 to 0.1. This impact is due to MannN representing channel roughness, which affects streamflow transit time and
volume.

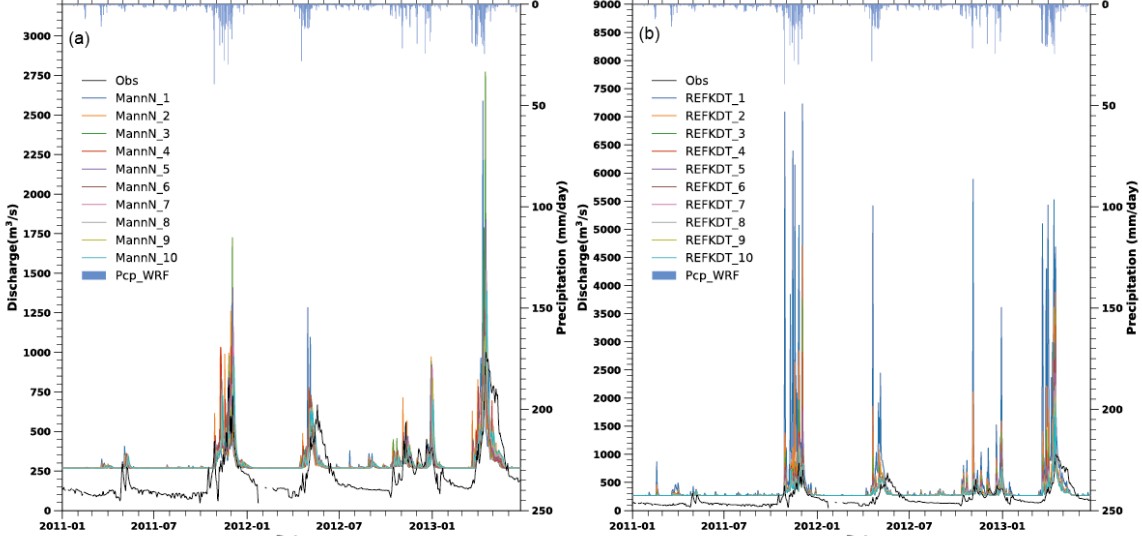


**Figure 6. The simulated WRF-Hydro discharge at Garissa from January 2011 to June 2013 from Manning roughness parameter (MannN)**
**and runoff infiltration coefficients (REFKDT) sensitivity tests, against the observation (Obs). MannN (or REFKDT) test consists of ten**
**simulations, with the MannN (or REFKDT) ranging from a near-zero (or 0.02) scale in MannN_1 (or REFKDT_1) experiment to a scale**
**of 4 (or 1) in MannN_10 (or REFKDT_10) with nearly equal intervals throughout. Precipitation from the WRF simulation (Pcp_WRF)**
**is shown at the top.**
Similarly, the REFKDT parameter also significantly impacts peak discharge in response to heavy rain. An increase in REFKDT
generally results in decreased discharge (Fig. 6 b and Table S4). Specifically, when the REFKDT scaling factor changes from 0.02
(REFKDT equals 0.1) to 1 (REFKDT equals 5), the peak-flow decreases from 7229 m$^3$ s$^{-1}$ to 1092 m$^3$ s$^{-1}$. In the WRF-Hydro
modelling system, the REFKDT parameter governs surface infiltration by partitioning runoff into the surface and subsurface
components (Schaake et al., 1996), meaning a higher REFKDT value allows more water into the subsurface, therefore reducing
surface runoff and peak discharge.

However, both MannN and REFKDT have minimal effects on alleviating the underestimation of dry-season flow in the above
WRF-Hydro simulations with the lake/reservoir module (LakeRaw), which remains largely unchanged despite variations of the
two parameters.

#### 4.2.4. Sensitivity analysis from groundwater components


Overall, adjusting groundwater component options could slightly alleviate the overestimation of dry-season flow (Fig. 7 and Table
S5). The dry-season flows from the two experiments all remain large overestimation with a considerable bias of 122 (81 %) and





161 (107 %) m³ s⁻¹. However, among the two experiments, the simulated discharge fluctuation in the
GWBASESWCRT_Passthrough experiment aligns better with the observation, compared to the GWBASESWCRT_Sink
experiment. The correlation coefficient ($r^2$) of the simuled discharge against the observation is 0.56 and 0.33 in
GWBASESWCRT_Passthrough and GWBASESWCRT_Sink experiment, respectively. The discrepancies in waveform led to an
earlier prediction of flood retreat. Given the enhanced performance of GWBASESWCRT_Passthrough experiment, we selected
the pass-through bucket module for the subsequent sensitivity analysis and calibration experiment.

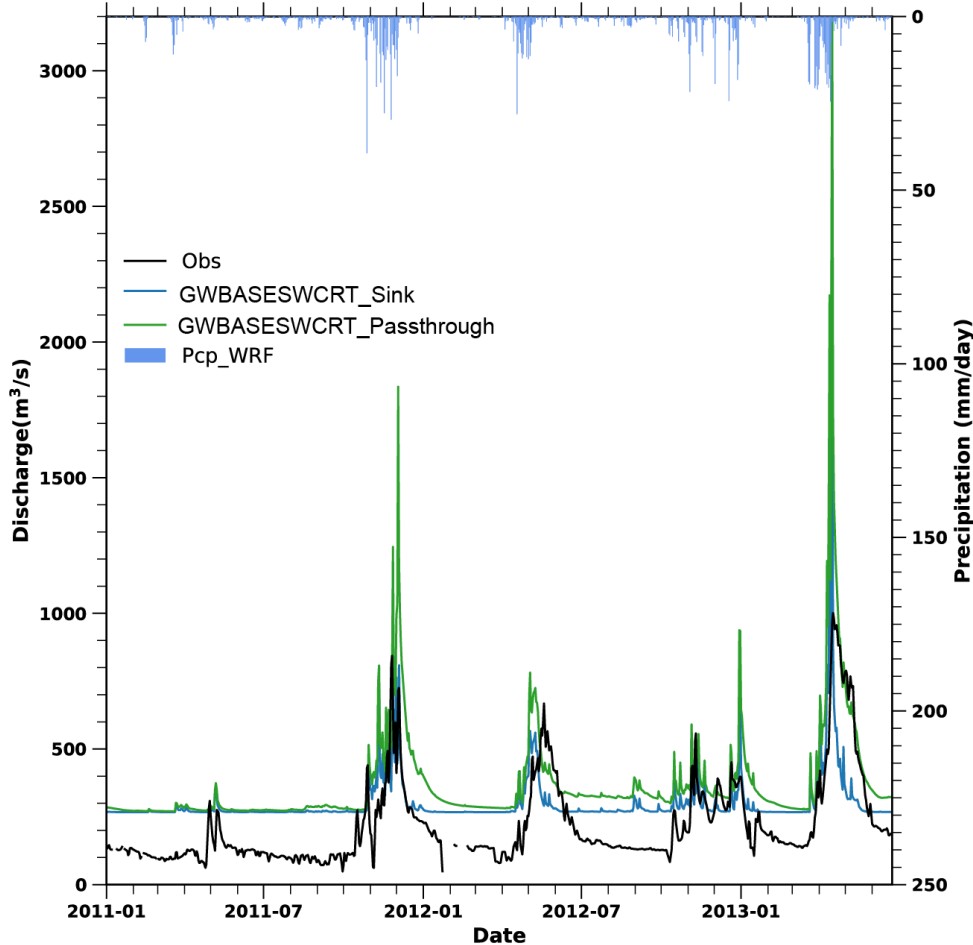


**Figure 7. The discharge evolution of the two experiments and the observation. One experiment creates a sink at the bottom of the soil**
**column, where water drains out of the system (GWBASESWCRT_Sink), while the other bypasses the bucket model and directly channels**
**all flow from the bottom of the soil column into the stream (GWBASESWCRT_Passthrough). Precipitation from the WRF simulation**
**(Pcp_WRF) is shown at the top.**

**4.2.5.    Sensitivity analysis from lake-related parameters**
From the Morris result (Fig. 8 and Table S6), lake-related parameters (i.e., LkMxE, WeirE, WeirC  WeirL,  OrificeA,  OrificeC,
and OrificeE) show a distinct influence on the discharge at Garissa. The overestimation of discharge was mitigated in the best
simulation with the largest NSE (the red line in Fig. 8 a). Among the eight lake-related parameters, the WeirE turns out to be the
most sensitive, as indicated by its top sensitivity rank (Fig. 8 b). Altering the WeirE from its maximum (maximum water level plus
half water depth) to its minimum (the default Orifice elevation) in the LakeRaw model with other parameters at their default (Table

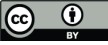



S6), resulted in an average discharge varying from 311 m³ s⁻¹ to 38 m³ s⁻¹, with model-data bias from 19 % to less than -85 %. This
sensitivity is particularly notable during the dry-season, causing a bias difference of 244 m³ s⁻¹ averaged in the dry season during
2011-2014, corresponding to -163 % of observations. This indicates that adjusting the lake-related parameters could alleviate the
overestimation of dry-season flow, showing potential to improve the model's performance. Notably, the eight parameters exhibit
distinct interdependence, as indicated by the large value of sigma/u (> 0.5) (Fig. 8 c), suggesting that parameter optimization should
be conducted globally rather than locally.

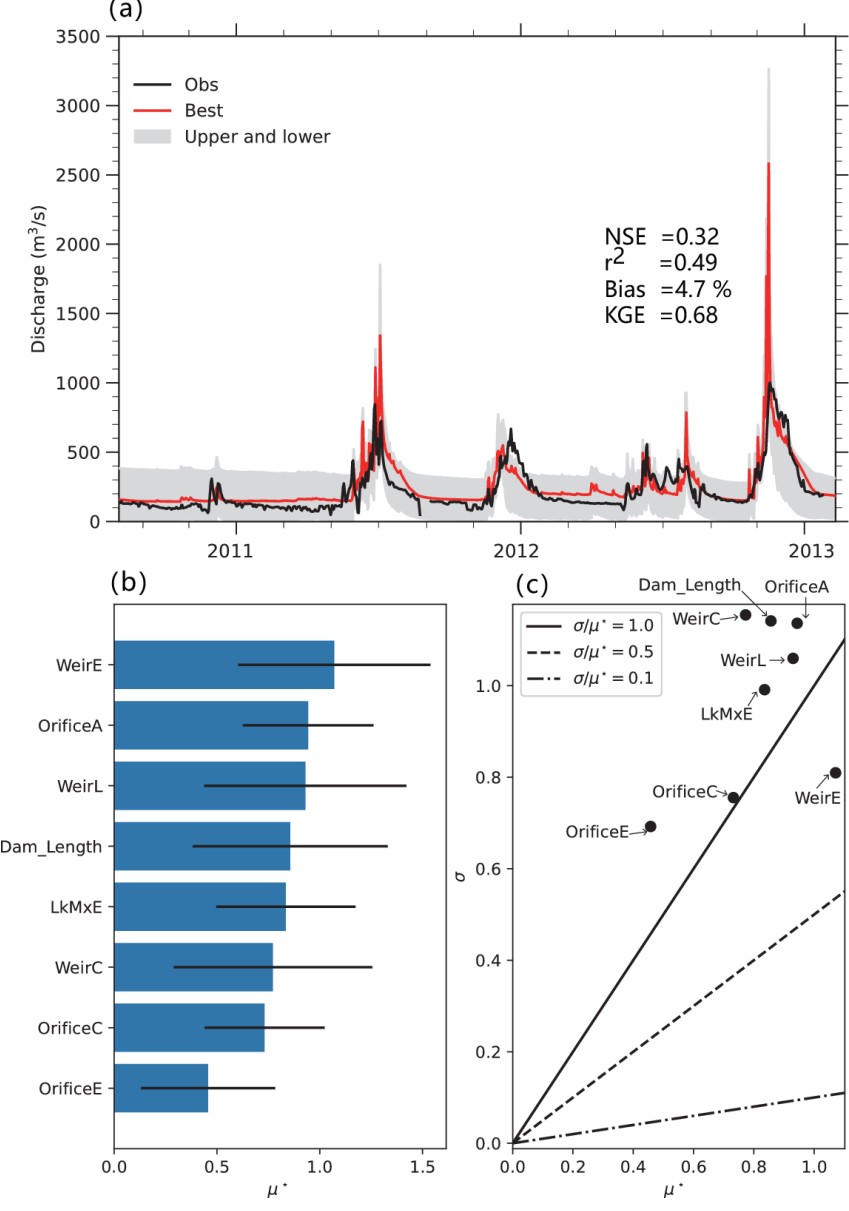


**Figure 8. The Morris result, including simulated discharge from 90 experiments against the observation (a), the sensitivity ranking (b)**
**and parameter interdependence (c). Nash-Sutcliffe Efficiency (NSE), coefficient of determination (r2), bias (Bias, unit: %), and Kling-**
**Gupta Efficiency (KGE) are calculated based on the best-simulated discharge at Garissa (with the largest NSE; shown in red) against**
**the observation. The u\* donates the sensitivity of a given parameter, with a higher value indicating greater sensitivity. The large value**



**of σ/u\* indicates stronger dependencies with other parameters. The sensitivity order is generated based on the model-data bias of the**
**simulated discharge at Garissa.**
Although adjusting lake-related parameters can alleviate the overestimation of dry-season flow, it induces another issue: the rain-
season flow discharge decreases synchronously, leading to its underestimation. Changes in the WeirE (in the LakeRaw modelling
with the other parameters as the default) cause rain-season flow from positive bias (52 m$^3$ s$^{-1}$, 19 %) to negative (-197 m$^3$ s$^{-1}$, -
71 %). This bias change is also observed in the peak-flow, which varied from an overestimation of 165 m$^3$ s$^{-1}$ (20 %) to an
underestimation of -127 m$^3$ s$^{-1}$ (-16 %). Fortunately, the rain-season flow underestimation could be re-tuned by REFKDT or MannN,
as well as the peak-flow.

Lakes with larger surface areas seem to play a dominant role in affecting discharge biases, as shown in Fig. S3. Adjusting
parameters for larger lakes, such as MASINGA, KAMBURU, and KIAMBERE, tends to cause greater variations, indicated by
larger standard deviations, compared to the small lakes, such as GITARU and KINDARUMA. Among the five lakes, MASINGA
(the largest, with an area of 111.6 km²) exhibits the most significant impact on discharge, with standard deviations of 21 % for
peak flow, 23.7 % for average discharge, 19 % for rain-season flow, and 34 % for dry-season flow. Conversely, KINDARUMA
(the smallest with an area of 2.1 km²) exhibits the least impact on discharge, with standard deviations of near zero (0.1 %, 0.3 %,
0.2 %, and 0.6 %), respectively.
**4.2.6.    The optimized results of WRF-Hydro modelling with lake/reservoir module**
Based on the sensitivity analysis result, we conducted a calibration involving the parameters outlined above, and the results are
shown in Fig. 9 and Table S2. Calibration of WRF-hydro modelling system with lake/reservoir module greatly improves the
simulation of river discharges in the TRB. The simulated discharge from LakeCal with a KGE of 0.70 and a bias of 9 %, is more
consistent with the observed flow process, compared to LakeRaw with a KGE of 0.35 and a bias of 40 %. The significant
overestimation of discharge in the LakeRaw (Sect. 4.2.1) model was notably reduced through the calibration of the lake/reservoir
module, although a slight overestimation still exists.



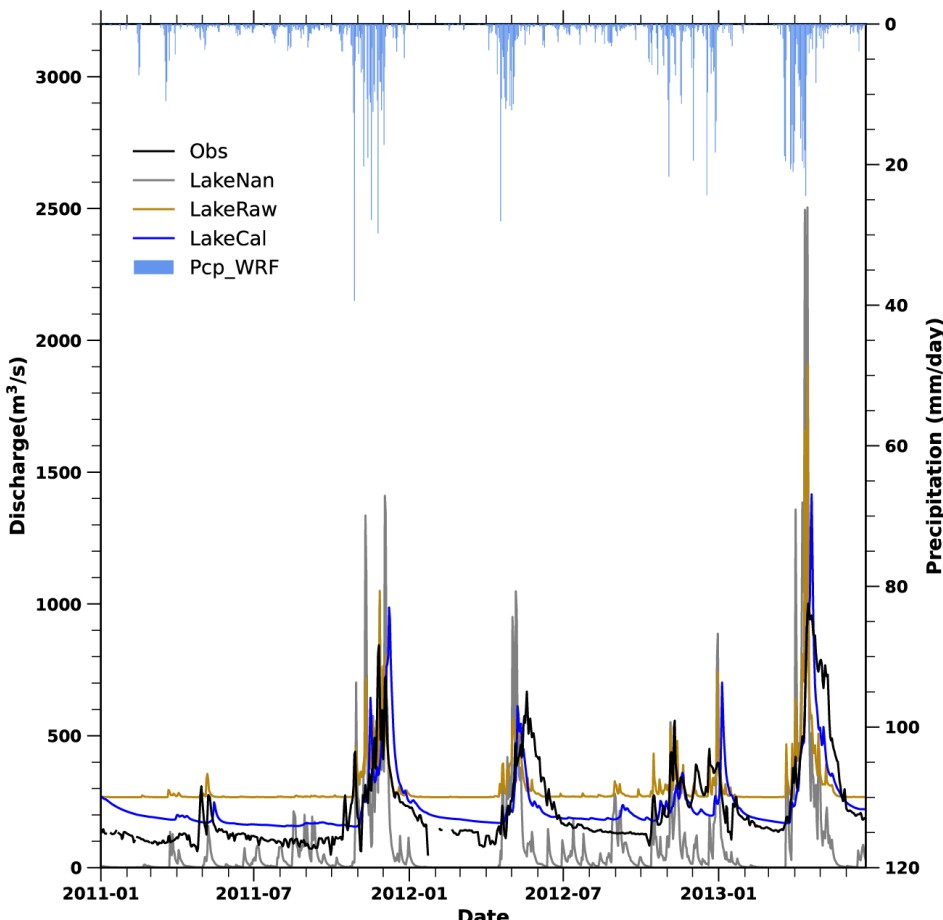

**Figure 9. The simulated discharges from three WRF-Hydro simulations against the observation. The three include WRF-Hydro simulation without lake/reservoir module (LakeNan in grey), WRF-Hydro with lake/reservoir module based on parameters from the LakeNan (LakeRaw, in brown) and the well-calibrated WRF-Hydro with lake/reservoir module (LakeCal, in blue). Precipitation from the WRF simulation (Pcp_WRF) is shown at the top.**

Notably, the modelling performance of WRF-Hydro simulation with the lake/reservoir module (LakeCal) is much better than that without lake/reservoir module (LakeNan). The KGE and bias are 0.16 and -53 in LakeNan simulation, in contrast to 0.70 and 9 % in LakeCal simulation. The improvement is especially for dry-season flow and peak-flow simulation, despite a slight overestimation of dry-season flow. The calibration of WRF-Hydro modelling system with lake/reservoir module corrects the overestimation of dry-season flow by 71 $m^3\ s^{-1}$, reducing the dry-season flow from 271 $m^3\ s^{-1}$(with a bias of 81 %) to 200.1 $m^3\ s^{-1}$ (with a bias of 34 %). Besides, the deviation in peak-flow, indicated by a bias of 174 % (144 $m^3\ s^{-1}$) decreased in LakeCal compared to the bias of 24 % (206 $m^3\ s^{-1}$) in the LakeRaw. Consistently, the overestimation of averaged discharge in both the dry-season and rain-season flow was reduced, with the bias changing from 81 % and 22 % to 34 % and -2 %. Due to this improvement in dry-season flow and peak-flow simulation, LakeCal better captures seasonal variation than the other two models. The $r^2$ is 0.75 in the LakeCal model, calculated over the monthly discharge against the observation, compared to 0.66 in the LakeNan simulation. Furthermore, the LakeCal could better capture the hydrograph shape during the rise and recession of floods, as indicated by the improved $r^2$ of 0.59, compared to 0.30 in the LakeNan and 0.33 in the LakeRaw. For example, during the MAM period in 2012 and 2013, the simulated onset and recession times of flooding by LakeCal were closer to the observed, than those from the LakeRaw and LakeNan. The earlier estimation of flood onset times in the LakeRaw was significantly alleviated in the LakeCal. The better




fit of the simulated discharge against the observation during flood rising and falling times in the WRF-Hydro system with
lake/reservoir module, indicates a promising ability to accurately forecast floods.
**4.3. Attribution of hydrological simulation enhancement**
The above skilled WRF-Hydro simulation driven by WRF precipitation (LakeCal, Fig. 9) could be attributed to the integration of
convection-permitting WRF simulation and the inclusion of lake/reservoir module. To qualify the contributions from CPWRF
simulation and lake module, we compared the well-calibrated WRF-Hydro simulation with lake/reservoir module driven by
CPWRF output (LakeCal) to the calibrated WRF-Hydro modelling without lake module forced by CPWRF output (LakeNan) and
the well-calibrated WRF-Hydro simulation with lake module driven by ERA5 (LakeCal-ERA5), shown in Figs. 9,10a and Table
S2.

The well-calibrated lake-integrated model forced by CPWRF output (LakeCal), outperforms both LakeNan driven by CPWRF
output and lake-integrated model forced by ERA5 (LakeCal-ERA5). Comparing LakeCal to LakeCal-ERA5, the WRF-improved
precipitation notably enhances the WRF-Hydro modelling performance, especially reducing the peak false (Fig. 10 a). The
simulation skill indicated by NSE, rises from 0.04 (LakeCal-ERA5) to 0.57 (the LakeCal) (Table S2), resulting in an NSE increase
of 0.53. Comparing the LakeCal to LakeNan, the inclusion of the lake/reservoir module significantly improves the WRF-Hydro
performance, distinct in alleviating under estimation of the dry-season flow and the overestimation of the peak flow. The NSE
rises from -1.10 (LakeNan) to 0.57 (LakeCal), which reflects an NSE increase of 1.67. Dividing by the total of the two increases,
improvements in hydrological simulation could be attributed 24 % (an NSE increase of 0.53) to WRF-refined precipitation and
76 % (an NSE increase of 1.67) to the inclusion of lake/reservoir module.

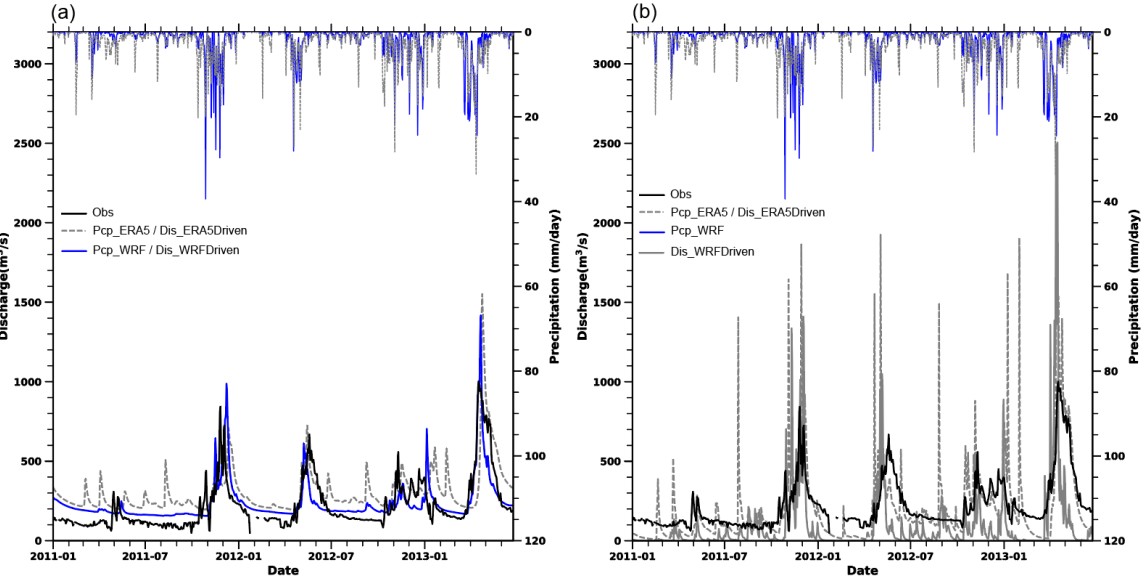


**Figure 10. The precipitations from WRF (Pcp_WRF, solid line on the top) and ERA5 (Pcp_ERA5, dash line on the top), as well as the**
**simulated daily discharge evolution from WRF-Hydro driven by WRF precipitation (solid line at the bottom colored blue in a and grey**
**in b) and ERA5-precipitation (dashed line at the bottom) against the observation (black dashed line). (a) and (b) indicate the results from**
**WRF-Hydro simulation with and without the lake/reservoir model, respectively.**



## 5. Discussion

### 5.1. Hydrological modelling improvement from convection-permitting WRF-simulated precipitation – Effect of precipitation forcing

Dynamic downscaling with refined resolution, especially in the convection-permitting scale, allows for a more reasonable representation of precipitation processes, particularly in mountainous areas (Schumacher et al., 2020; Li et al., 2020). The convection-permitting WRF simulation tends to improve local (e.g., mesoscale) scale processes and interactions between local and large-scales, especially over complex terrain (Kendon et al., 2021; Guevara Luna et al., 2020; Schmidli et al., 2006). Woodhams et al.'s research (2018) demonstrates that the convection-permitting WRF model shows greater skill than the global model, in particular on sub-daily time scales and for storms over land. It thus potentially contributes to added value in precipitation simulation. In our study, the WRF simulation improves the precipitation simulation (Sect. 4.1), especially, reducing the overestimation of light rainfall (1-15 mm day$^{-1}$) events compared to ERA5 (Fig. 3 and Table 8). Consequently, the hydrological simulation with the lake/reservoir module, using WRF precipitation as input (LakeCal), showed significant improvement, particularly in reducing false peak events, compared to that using ERA5 precipitation (LakeCal-ERA5) (Fig. 10a). This improvement related to peak flow is also evident in the WRF-Hydro simulation without the lake/reservoir module (Fig. 10b).

### 5.2. Hydrological modelling improvement from convection-permitting WRF-simulated precipitation – Effect of lake/reservoir module

The lake/reservoir module is crucial for improving hydrological simulations over TRB in East Africa. Possible factors contributing to the overestimation issues that can occur even with sufficient spin-up time. Factors such as the groundwater component, key hydrological parameters, and lake-related parameters. Despite some adjustments, the groundwater component (Sect. 4.2.4) and key hydrological parameters (Sect. 4.2.3) have a limited ability to alleviate the overestimation of dry-season flow in WRF-Hydro simulation without lake/reservoir module (LakeNan). In contrast, tuning lake-related parameters could significantly influence downstream discharge (Sect. 4.2.6). This underscores the important role of lake/reservoir module in enhancing hydrological simulations in the data scarcity regions that contain lakes or reservoirs.

Lake/reservoirs play a crucial regulatory role, storing water during the wet season (especially peak-flow) and releasing water during the dry season (Zajac et al., 2017; Hanasaki et al., 2006). In our study, hydrological simulations without lake/reservoir module in the TRB, which includes five lakes, show significant underestimation (-78 %) in dry-season flow and overestimation (24 %) in peak-flow. These biases (dry-season flow underestimation and peak-flow overestimation) are common issues in East Africa, as highlighted by Arnault et al., (2023). Previous studies demonstrated that enhancing reservoir hydrological processes can improve simulation accuracy (Hanasaki et al., 2006; Lehner et al., 2011) for basins with reservoirs or lakes. Our results confirm that the well-calibrated WRF-Hydro system with the lake/reservoir module significantly reduces the underestimation of dry-season flow and overestimation of peak-flow. The lake/reservoir module helps to correct the underestimation of dry-season flow, adjusting the dry-season flow bias from -78 % in LakeNan simulation to 34 % in the LakeCal, despite some positive bias. Additionally, the peak-flow bias in the lake/reservoir simulation decreased to 17 %, compared to the value of 24 % in LakeNan simulation.

### 5.3. Uncertainties of the hydrological modelling





Although the lake module improves WRF-Hydro simulation, the model expressed as a water balance equation with a simple level-pool scheme could induce uncertainties in the hydrological simulation, due to the insufficient physical mechanism, lack of consideration for human activities and small tributaries in the upstream of lakes. For example, lake water levels may be not well presented (Fig. S4 and Table S6). In the LakeCal simulation, the water level devotion can reach -191m (-28 % of the water level observation averaged over 2011-2015) at KIAMBERE. Moreover, the water level fluctuations between the simulations and observation show large differences, with $r^2$ ranging from near zero (0.005) to 0.25 for the five lakes.

The groundwater component may cause uncertainties, as we used a pass-through bucket module that directs all flow from the soil column into the channel without recharging groundwater. This approach might not capture the intermittent groundwater recharge from seasonal rainfall in the TRB (Taylor et al., 2013). This leads to potential inaccuracies in simulating groundwater processes and their interaction with surface water in East Africa.

The benchmark data in the data scarcity area (East Africa) presents challenges for model evaluation. For example, uncertainty from IMERG precipitation over East Africa (Dezfuli et al., 2017), may complicate precipitation evaluation. WRF precipitation shows an underestimation of extreme precipitation (i.e. 90-100 quantiles) against the IMERG (Fig. 3), while the simulated discharge from LakeCal driven by WRF-precipitation does not show a distinct underestimation of extreme flow (against the observation) as expected (Fig. 10b). The absence of the underestimation of extreme flow suggests a potential overestimation of extreme precipitation from IMERG against the real. The overestimation of IMERG precipitation in Africa has been demonstrated in previous research (Maranan et al., 2020; Dezfuli et al., 2017), which consequently creates the illusion of some underestimation in WRF precipitation (Fig. 2 and Table 7). Such erroneous underestimation of WRF precipitation was also indicated by the general overestimation of extreme flow in LakeCal simulation (Fig. 10 a-b).

Future work will focus on refining the hydrological simulation over East Africa with an advanced dynamical lake/reservoir module (Wang et al., 2019) and an enhanced groundwater component. Bias correction of hydrological output variables could also be considered to improve the hydrological simulation (Tiwari et al., 2022). Besides, reliable benchmarks in East Africa will be crucial for evaluating WRF simulation performance.

## 6. Conclusion

In this article, we presented a seamless, consistent meteorological-hydrological modelling system for hydrological simulation in East Africa. The hydrological simulation is enhanced by CPWRF and lake module, through a case study in the TRB.

(1) The refined precipitation from CPWRF simulation improves the hydrological simulation, which makes an NSE increase of 0.53 when comparing LakeCal to LakeCal-ERA5, contributing to a 24 % enhancement in the hydrological simulation. The CPWRF simulations produce more accurate precipitation estimates than ERA5, particularly for the precipitation amount over mountainous regions and light precipitation events (1-15 mm day$^{-1}$) in the dry seasons (JF and JJAS). The well-calibrated lake-integrated simulaiton driven by CRWRF output (LakeCal) was improved especially alleviating the peak false, compared to that by ERA5 (Lake-ERA5).

(2) Additionally, the incorporation of the lake/reservoir module in the WRF-Hydro system mitigates the bias of dry-season flow and peak flow when comparing LakeCal to that without lake (LakeNan), with an NSE increase of 1.67, contributing to a 76 %



improvement in hydrological simulation. The lake module could distinctly affect discharge through lake-related parameters. The lake module makes river discharge more sensitive to spin-up time, which prolongs the spin-up time required for the streamflow simulation to achieve stability, with dry-season flow exhibiting higher sensitivity compared to the rain-season flow. Adjustments to the lake-integrated model's parameters (runoff infiltration rate, Manning's roughness coefficient, and the groundwater component) have minimal impact on he dry-season flows.

Our study marks the improved streamflow simulation using WRF-Hydro modelling system by integrating with a lake/reservoir module and convection-permitting WRF simulation. This approach offers a promising tool for conducting reliable hydrological simulations in data-scarce regions of East Africa. Previous studies have rarely addressed the sensitivity analysis and parameter tuning of the lake/reservoir module within the WRF-Hydro system. Our findings offer new insights into the impacts of lake/reservoirs on hydrological simulations, providing valuable benchmarks for optimizing hydrological modelling, especially those involving lake/reservoir components.

Utilizing the lake/reservoir module and convection-permitting modelling, our approach could address some of the challenges related to flood/drought simulation uncertainty and lay the groundwork for more sophisticated hydrological modelling related to more complex water cycles. This enhancement from the approach has the potential for more accurate flood and drought predictability, facilitating more informed decision-making in water resource management, as well as flood and drought risk mitigation. Ultimately, this supports sustainable environmental stewardship in regions susceptible to hydrological variability and change.

## Acknowledgments

This research was supported by the European Union's Horizon 2020 research and innovation program under grant agreement no. 869730 (CONFER), National Natural Science Foundation of China (Grants 42205057 and 42125502), Project funded by China Postdoctoral Science Foundation (Grant 1232192), the German Science Foundation (DFG) project: Large-Scale and High-Resolution Mapping of Soil Moisture on Field and Catchment Scales Boosted by Cosmic-Ray Neutrons (COSMIC-SENSE, FOR 2694, grant KU 2090/12-2). The computer resources were available through the RCN's program for supercomputing (NOTUR/NORSTORE); projects NN9853K and NS9853K. Thanks to ChatGPT for improving the language.

## Code availability

WRF code is available from https://github.com/wrf-model/WRF. WRF-Hydro code is available from https://github.com/NCAR/wrf_hydro_nwm_public.

## Data availability

All WRF-Hydro simulation data in this paper are available from the authors upon request (lingzhang@cug.edu.cn and luli@norceresearch.no).

## Competing interests



The authors declare that they have no conflict of interest.
**Author contribution**
Ling Zhang and Luli developed the idea together and designed the sensitivity experiments. Ling Zhang perfected the idea, carried
model run, data analysis, and prepared the original manuscript and the subsequent modificaiton. Joël Arnault, Lu li, and Anthony
Musili Mwanthi designed the WRF/WRF-Hydro model. Zhongshi Zhang, Joël Arnault, Stefan Sobolowski, Pratik Kad contributed
the review & editing. The other co-authors (Mohammed Abdullahi Hassan, Tanja Portele, Harald Kunstmann) provided
suggestions related to flood simulation, which facilitated the work.

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
