# Peer review of "Enhanced hydrological modeling with the WRF-Hydro lake/reservoir"

_Hydrology and Earth System Sciences, 2024_

## Author Comment (AC2)

Dear Editor and Reviewer,

We sincerely thank the Reviewer for the constructive comments. Your recommendations are valuable and helped enhance the analysis and sharpen the argumentation in the manuscript. In the following, we address each comment. The reviewers' comments are presented in black, and our responses are **in blue**.

**Overview of the paper**

The paper presents a hydrological simulation in a data-scarce region, utilizing a convection-permitting climate model and a lake-integrated hydrological model. It highlights and quantifies the improvements brought by convection-permitting WRF simulations and the inclusion of lake and reservoir modeling. A key contribution of the paper is its identification of specific areas of improvement, particularly at peak and low-flow points, and a detailed explanation of the underlying causes and attributions of these improvements.

The study addresses the added value of convection-permitting modeling in hydrological simulations and the integration of the lake/reservoir module within WRF-Hydro, both of which are topics that have been rarely explored. The findings provide fresh insights into the benefits of using a convection-permitting climate model and the lake/reservoir module, offering valuable benchmarks for optimizing hydrological modeling, especially in regions with limited data availability. As noted by the authors, this enhanced hydrological simulation has potential future applications in forecasting extreme hydrological events, such as floods and droughts.

The paper aligns well with the scope of HESS, as it focuses on hydrological simulation, especially in a data-scarce region. The use of the convection-permitting climate model and lake-integrated hydrological model addresses critical concerns in the climate-hydrology field. Furthermore, the study area, East Africa, is particularly relevant due to the scarcity of data, the complexity of hydrological simulations, and the frequent occurrence of extreme flood and drought events.

In conclusion, this paper offers significant value and is suitable for publication in HESS. However, I have several suggestions and comments for consideration, as outlined below.

Major comments

Although this manuscript is well-structured and comprehensive, with some well-written sections, it requires careful editing by professional English editors. Special attention should be given to sentence structure, as well as minor spelling and grammatical errors, to ensure that the

study's goals and results are clear to the reader.

Reply: Thanks for your comment. We have revised the English language throughout the manuscript to enhance clarity and readability.

The conclusions should be presented with caution. The sensitivity of the simulated peak flows to the spin-up time, based on a single event in 2011. The conclusion may vary if different regions or other peak flow events are considered. I recommend adding further discussion on this point.

Reply: Thank you for the suggestion. We added further discussion on this point in Sect. 5.4. The added discussion now reads as follows.

"Also, uncertainty may exist in the sensitivity analysis of the simulated peak flow to spin-up time, which was based on a single event (the largest observed peak from 2010 to 2014) at a specific discharge station (i.e., Garissa or Rukanga). The conclusion, especially about the spin-up time required for model stabilization, may vary when different regions or other peak flow events are considered. For example, a WRF/WRF-Hydro simulation (Lu et al., 2020) exhibits that initialization times needed for soil moisture stabilization differ for different basins in Western Norway. The varying spin-up periods required for flow stabilization between the dry and rainy seasons (Sect. 4.2.2 and Fig. 5 d-e) indicate the possible sensitivity of peak flow to spin-up duration across different peak flow events. The sensitivity of different regions and other peak flow events to spin-up time will be further investigated."

The authors should verify the model configuration. Figure 1 indicates that WRF is directly driven by ERA5, but lines 156-157 suggest a nested simulation domain. Please double-check this for accuracy.

Reply: Thanks for your careful review. We have corrected it in the revised manuscript (Line 163) as below.

"The convection parameterization was turned off for the WRF simulation,…".

When evaluating the WRF simulation, the authors focus primarily on bias. However, a smaller bias does not necessarily indicate a better simulation. I suggest using additional indices, such as a Taylor diagram, to support the conclusions.

Reply: Thanks for your valuable suggestion. We acknowledge that focusing only on bias may not fully capture the simulation's accuracy. We have added additional evaluation indices, including the Taylor diagram, to provide a more comprehensive assessment of the WRF simulation's performance. To ensure the continuity of the context, we have revised Section 4.1 as follows.

"4.1.     WRF Precipitation refinement

[revised manuscript text omitted]

Additionally, we have updated the methodology in Sect. 3.4 to include details on the Taylor Diagram (Line 312-314), which reads as follows.

"To fully evaluate the simulated precipitation by CPWRF, we also employed Taylor diagrams (Taylor, 2001), which present a concise statistical summary in terms of spatial correlation (indicated by correlation coefficient), and spatial variance (indicated by normalized standardized deviation). A higher spatial correlation and a spatial variance closer to one indicate better simulation skills."

We also add discussion about the performance of the model in the Uncertainty, which reads as follows.

"Different metrics (r, bias, and normalized standardized deviation) were used to provide a more comprehensive assessment of the CPWRF's performance, which may cause contradictory or different evaluations of its skill. Each metric emphasizes different aspects of model performance and leads to divergent conclusions about the model's strengths or weaknesses. For instance, seasonal precipitation from the CPWRF result exhibits apparent added value to the forcing data over mountainous areas (Fig. S2 a-e), which however is not distinct in the Taylor figure (Fig. S1). This discrepancy arises because the region with apparent added value is mainly centered on Mount Kenya, whereas the mountainous region in the Taylor diagram analysis includes areas above 1600 m, extending beyond Mount Kenya. Therefore, further in-depth

research is needed to fully assess the performance of CPWRF with these different metrics and explain the possible discrepancy."

The manuscript emphasizes the importance of CPM in East Africa. If the authors add a discussion of the added values of CPM with respect to its driving forces, this manuscript would be more informative.

Reply: Thank you for this insightful suggestion. To address this, we have expanded Section 5.2 including the added value of Convection-Permitting Models (CPM) in East Africa, particularly in relation to their driving forces. This additional content emphasizes CPM enhances the representation of convective processes compared to coarser-resolution models, over Mount Kenya and the surrounding area. We believe this provides a more thorough understanding of CPM's benefits in this region. To ensure the continuity of the context, we have revised Section 5.2 as follows.

"**5.2.  Hydrological modeling improvement from precipitation simulated by convection-permitting WRF – Effect of precipitation forcing**

Dynamic downscaling at convection-permitting resolution allows for a more accurate representation of precipitation processes. The convection-permitting WRF simulation enhances local (e.g., mesoscale) processes and interactions between local and large-scales, especially over complex terrain (Kendon et al., 2021; Guevara Luna et al., 2020; Schmidli et al., 2006; Schumacher et al., 2020; Li et al., 2020). As a result, CPWRF potentially contributes to improving precipitation simulation in our study (Sect. 4.1), especially reducing bias in seasonal precipitation over mountainous areas, and light rainfall (1-15mm day-1) probability in the dry season compared to ERA5 (Fig. 3 and Table 8).

The improvement in the seasonal precipitation over mountainous regions and rainfall probability can be supported by the spatial distribution of the added value (AV) in seasonal precipitation with respect to the driving forces (Fig. S2). The WRF simulation adds consistent value to ERA5 over the mountainous areas across all four seasons (MAM, OND, JF, and JJAS). The area with positive AV is mainly over Mount Kenya and its surrounding areas, with the positive AV being particularly distinct during the dry season. WRF also adds value to ERA5 in the light rainfall probability (Fig. S2 f-k), as demonstrated in Sect. 4.1. The basin averaged AV of CPWRF over the probability of light precipitation events are 0.32, 0.26, 0.30, and 0.07 in MAM, OND, JF, and JJAS, respectively. The positive AV of WRF with respect to ERA5 over the extreme rainfall probability, also concentrates around Mount Kenya, consistently across all four seasons (Fig. S2 l-p). Previous studies (Giorgi et al., 2022) have demonstrated that the added value of WRF simulations is influenced by various factors, including timescale, variables, regions, and uncertainty of the benchmark. Therefore, further in-depth research is required for a more reliable AV assessment.

Due to the precipitation improvement from WRF, hydrological simulation with WRF precipitation as a driving force (LakeCal), showed significant improvements, compared to simulations driven by ERA5 (LakeCal-ERA5) (Fig. 10a). These improvements are particularly notable in reducing false peak simulations, likely due to the reduction in the overestimation of light rainfall probability. The enhancement in peak flow simulation is also observed in the WRF-Hydro model without the lake/reservoir module (Fig. 10b)."

Besides, we have also added a description of the AV (added value) methodology in Sect. 3.4, which reads as follows.

"3.4.    Evaluation of simulated precipitation from CPWRF

To assess whether the CPWAR has advantages over their driving forces (ERA5), added value (AV) proposed by Dosio et al. (2015) was applied, expressed as follows.

$$AV = \frac{(X_{ERA5} - X_{IMERG})^2 - (X_{CPWRF} - X_{IMERG})^2}{\max\left((X_{ERA5} - X_{IMERG})^2, (X_{CPWRF} - X_{IMERG})^2\right)} \tag{3}$$

$X_{ERA5}$, $X_{CPWRF}$, and $X_{IMERG}$ indicate precipitation from the driving forces (ERA5), CPWRF simulation, and benchmark (IMERG), respectively. The added value (AV) from CPWRF is defined as the performance difference between itself and the driving forces for precipitation in a specific region and period. If the CPWRF adds value to the driving forces (ERA5), the AV is positive, whereas a negative AV suggests no adding value.

To fully evaluate the simulated precipitation by CPWRF, we also employed Taylor diagrams (Taylor, 2001), which present a concise statistical summary in terms of spatial correlation (indicated by correlation coefficient), and spatial variance (indicated by normalized standardized deviation). A higher spatial correlation and a spatial variance closer to one indicate better simulation skills."

Thank you once again for your suggestion. We believe these additions enhance the manuscript by providing a more comprehensive view of CPM's contributions in East Africa.

Minor comments

Line17-18: Replace "the upper and middle stream of the Tana River basin was" with "the upper and middle streams of the Tana River basin were ".

Reply: Thanks and done.

Line19-20: This sentence is ambiguous. Please revise it.

Reply: Thank you for pointing it out. We have revised the sentence and changed it to "We

performed convection-permitting (CP) simulations using the Weather Research and Forecasting (WRF) model and conducted lake/reservoir-integrated WRF Hydrological modeling (WRF-Hydro) driven by CPWRF output.".

Line21: Replace " using IMERG as the benchmark" with " when benchmarked against IMERG

Reply: Thanks and done.

Line22-23 Change "alleviates the peak false" to "alleviates the false peak simulation".

Reply: Thanks and done.

Line24: Change "NSE" with "NSE (Nash-Sutcliffe Efficiency)".

Reply: Thanks and done.

Lin29-30: There are two terms "lake" and "lake/reservoir" which seem to represent the same thing. It would be better to unify them for consistency throughout the document.

Reply: Thanks and done.

Line29-30: Replace "highlight the enhanced hydrological modelling capability with" with "highlight the enhanced capability of hydrological modelling using".

Reply: Thanks and done.

Line123: Replace "resulting" with "which results".

Reply: Thanks and done.

Line129: Please replace "S 1.25°~N 0.50°" with "S 1.25°-N 0.50°".

Reply: Thanks and done.

Line139: Replace " the upper and middle stream of the Tana River Basi" with " the upper and middle streams of the Tana River basin".

Reply: Thanks and done.

Line:154: Usually, one month spin-up is sufficient for WRF downscaling.

Reply: Thanks for pointing it out. We have revised Sect. 3.3 as the follows.

"To obtain convection-permitting modeling precipitation, we used the Advanced Research WRF (WRF-ARW) model of version 4.4 (Skamarock et al., 2019) with the designed domain of 5 km spatial resolution (Fig. 1). The lateral boundaries were forced with the 6-hourly ERA5 reanalysis with a spatial resolution of 0.25 degrees (Hersbach et al., 2020). The model was set with 50 vertical levels up to 10hPa. The convection parameterization was turned off for the WRF simulation, the Mellor-Yamada Nakanishi Niino Level 2.5 (MYNN2.5) Scheme

(Nakanishi and Niino, 2006) for the planetary boundary layer, the RRTM scheme for longwave radiation (Mlawer et al., 1997), and the Dudhia Shortwave scheme for shortwave radiation (Jimy Dudhia, 1989). The Noah-MP Land Surface model ('Noah-MP LSM', Yang et al., 2011) was used for the land surface scheme.

The model runs from 1 January 2010 to 31 December 2014. Typically, WRF simulations require a spin-up of about one month, which should ideally be excluded from precipitation evaluation. However, given the limited length of simulated precipitation, the subsequent analysis is based on full precipitation simulation from January 2010 to December 2014."

Line182-183: Please add "(with the lake/reservoir module inactive)" behind "without the lake/reservoir module".

Reply: Thank you for the suggestion. We have revised it as requested.

Line185: Replace "of the Garissa discharge." with "of simulated discharge against the observation at Garissa".

Reply: Thank you for the suggestion. We have revised it.

Line208-210: Replace "which may affect the subsequent sensitivity analysis and hydrological modelling assessments." with "which may potentially affect the result of subsequent sensitivity analyses and the performance of the hydrological simulation."

Reply: Thank you. We have revised it.

Line260: Replace "For each lake test" with "For each test".

Reply: Thanks and done.

Line274: Replace "Each lake" with "Each".

Reply: Thanks and done.

Line306-307: Please change the unit "mm a$^{-1}$" to "mm". The unit should be corrected in the whole text.

Reply: Thanks and done.

Line368: Replace "GIS pre-41.processing" with "GIS pre-processing".

Reply: Thanks and done.

Line408: Replace "demonstrate" with "demonstrates".

Reply: Thanks and done.

Line411: The view "a larger lake seems to require more time to reach equilibrium." depends.

You should add some references.

Reply: Thanks for the helpful suggestions. We did not find a related reference. So, we have removed this sentence and revised the related paragraph as follows.

"The water levels from the lake/reservoir-integrated model show a consistent spin-up period of 4 years across nearly all five lakes for the entire period, as well as during both the rainy and dry seasons (Fig. S2). Although KIAMBERE (one of the five lakes) exhibits a spin-up period of 3 years during the rainy season (Fig. S2e), it can be considered nearly 4 years due to the uncertainty in determining the spin-up time required for the stabilization of specific variables. Since the lakes are interconnected, the stabilization time is governed by the longest spin-up period. This may result in nearly the same initialization time for all five lakes (Table 1)."

Line499: Replace "-53" with "-53%".

Reply: Thanks for pointing out our careless, and done.

Line527: Replace "under estimation" with "underestimation".

Reply: Thanks and done.

Line529: "WRF-refined precipitation" is not idiomatic. Please revise it.

Reply: Thanks and done.

Line544: Replace "Woodhams et al.'s research (2018) demonstrates" with "Woodhams et al. (2018) demonstrates".

Reply: Thanks and done.

Line554: Delete ". Factors".

Reply: Thanks and done.

Line565: Replace "Arnault et al.," with "Arnault et al."

Reply: Thanks and done.

Line574-575: Replace " lake water levels may be not well presented" with "it shows a limited skill for simulating water level".

Reply: Thanks and done.

Line577: Please replace " with r2 ranging from near zero (0.005) to 0.25 for the five lakes." with " with r2 of the simulated discharge against the observation at Garissa less than 0.25 for all the five lakes."

Reply: Thanks and done.

Line599: Replace "a seamless, consistent meteorological-hydrological modelling system" with "a seamless and consistent meteorological-hydrological modelling system".

Reply: Thanks and done.

Line601-602: The sentence "which makes an NSE increase of 0.53 when comparing LakeCal to LakeCal-ERA5" is not clear. Please correct it.

Reply: Thanks for pointing it out. We have revised the text as "which result in an increase of 0.53 in NSE between the well-calibrated lake-integrated WRF-Hydro simulation driven by CRWRF output (LakeCal) and that driven by ERA5 precipitation (LakeCal-ERA5)".

We thank the reviewer for the valuable comments, which have helped improve our manuscript. We hope the revisions meet your expectations and strengthen our study.

---

## Author Response (AR1)

Dear Editor and Reviewer,

We sincerely thank the reviewers for the constructive comments. Your recommendations are valuable and helped enhance the analysis and sharpen the argumentation in the manuscript. The reviewers' comments are presented in black, with our replies **in blue**.

**Response to Reviewer #1:**

**Overview of the paper**

This manuscript presents an enhanced hydrological simulation using a resolution-refined climate model coupled with a more comprehensive hydrological framework. The improvements achieved through convection-permitting WRF simulations and the integration of lake and reservoir modeling are evident, particularly in addressing biases in peak flow and dry-season flow during discharge simulations. These findings emphasize the enhanced capability of hydrological simulations when using refined climate models and lake-integrated approaches.

The study focuses on two key topics: the added value of convection-permitting models and the integration of hydrological models with lake systems. This work holds great value in addressing challenges related to unreliable hydrological simulations, especially over Eastern Africa. It provides critical benchmarks for optimizing hydrological modeling, contributing to improved flood and drought forecasting and loss reduction in water management applications.

In my opinion, this paper gives an incremental advancement in the hydrological modelling field more than novelty, considering the convection-permitting modelling and lake-integrated hydrological simulation. The paper is well written, and quite comprehensive and well-structured. The topic is of interest and fits the journal scope, I suggest a few major revisions and some minor before publication in HESS. I believe a minor revision is needed and the revised manuscript will be much better. My comments are listed below.

**Both Major comments 1 and 2, as well as the minor comment Line154, are related to the time span for model simulation or evaluation.**

Reply: While we acknowledge that conducting WRF simulations over a longer period (preferably more than 10 years, instead of the 5 years from 2010 to 2014) would be ideal, we are constrained by the fact that WRF/WRF-Hydro simulations are highly

time-consuming and resource-intensive. Given that the project has recently ended, additional computational resources or time to perform these extensive simulations are no longer available. Consequently, our study is based on the existing WRF simulation timeframe.

Moreover, previous studies have shown that WRF/WRF-Hydro simulations are acceptable for regions like Africa, even with timeframes shorter than five years. Zhang et al. (2024) conducted WRF/WRF-Hydro simulation over 2015-2018 to assess how soil hydrophysical properties influence regional land-atmosphere coupling and the water cycle (involving precipitation) over the southern Africa region. Quenum et al. (2022) employed WRF/WRF-Hydro Modeling over 2008-2010 to explore the abilities of the fully coupled WRF-Hydro modeling system to simulate discharge and precipitation in the Ouémé River in West Africa.

Given the limited length of WRF simulation, the paper conducts precipitation evaluation and analysis based on data from January 2010 to December 2014. Typically, WRF simulations require a spin-up period of about one month (Minor comment Line 154), and the first month (January 2010) should be excluded from the evaluation of precipitation simulations. However, the precipitation evaluation period remains constrained, and the subsequent analysis focuses primarily on the rainy seasons (MAM and OND) and extreme events. Therefore, we employed simulated precipitation data from January 2010 to December 2014, without eliminating January 2010.

For the hydrological simulation based on the WRF-Hydro modelling, driven by the WRF output for 2010-2014, a spin-up time of at least one year is necessary (Figure 5). Therefore, our study WRF-Hydro simulations cover 2011-2014, with 2010 as the spin-up.

In future work, if time and resources permit, longer-term WRF simulations, along with WRF-Hydro simulations and data analysis, will be considered to reduce uncertainties.

References:

1. Zhang, Z. et al. Sensitivity of joint atmospheric-terrestrial water balance

simulations to soil representation: Convection-permitting coupled WRF-Hydro simulations for southern Africa. Agricultural and Forest Meteorology 355, (2024).

2. Quenum, G. M. L. D. et al. Potential of the Coupled WRF/WRF-Hydro Modeling System for Flood Forecasting in the Ouémé River (West Africa). Water (Switzerland) 14, (2022).

**Major comments**

1. The data for precipitation evaluation is from 2010 to 2014, while the data for discharge is from 2011 to 2014. I suggested using the same time series (i.e. 2010-2014) for precipitation and discharge evaluation.

Reply: Thanks for your suggestion. While we should use the same time series for precipitation and discharge analysis, different spin-up times could be acceptable due to the limited length of the simulation. For precipitation with a limited time length, we should include 2010 and thus conduct precipitation analysis based on the data over 2010-2014. However, WRF-Hydro simulation requires a full year of spin-up, so discharge evaluation excludes the first year, and thus is based on 2011-2014. While we acknowledge this mismatch, we believe that different spin-up times could be acceptable. We have added the clarification in the revised manuscript (Line 283-285). Please see our detailed explanation at the beginning of the replies.

The added part is as follows.

"Typically, we should use the same time series for discharge analysis as for the precipitation evaluation (2010-2014). However, since WRF-Hydro requires at least one year of spin-up, the discharge evaluation excludes the first year, focusing instead on the period from 2011 to 2014." (Line 283-2855)

2. If it is possible, longer time period of data for precipitation evaluation is necessary. Using data over 2010-2014 to calculate the monthly precipitation is not enough, which is usually more than 10 years.

Reply: We appreciate the comment. Please see our answer at the beginning of the replies.

3. It is not easy to understand how to calculate the attribution of discharge changes from

WRF-refined precipitation and lake-integrated WRF-Hydro model in Sect. 4.3. Please add it to the method section.

Reply: Thank you for the comment. We have added it in Sect. 3.5 of the methodology in the revised manuscript. Please see below.

"**3.5.    Attribution of hydrological model improvement to convection-permitting WRF simulation and lake/reservoir module**

To assess the contributions of CPWRF simulations and the lake/reservoir module, we compared three models: (1) the calibrated WRF-Hydro model without the lake/reservoir module, driven by CPWRF output (LakeNan), (2) the well-calibrated WRF-Hydro model integrated with the lake/reservoir module, also driven by CPWRF output (LakeCal), and (3) the well-calibrated WRF-Hydro simulation with the lake/reservoir module, driven by ERA5 (LakeCal-ERA5). We calculated the NSE value of simulated discharge against observed data for each model. Next, we computed the NSE increment between LakeCal relative to LakeNan representing improvements due to CPWRF precipitation, and the increment between LakeCal and LakeCal-ERA5 reflecting the influence of the lake/reservoir module. The ratio of CPWRF precipitation-induced or lake/reservoir module-induced NSE increment to the total increment is provided as the attribution of hydrological simulation improvements to the CPWRF simulations or the lake/reservoir module." (Line 303-313)

4. The Sect. 4.3 should be put at the front of the Discussion, since it explains why the hydrological simulation improves.

Reply: Thank you for the suggestion. We have now moved Section 4.3 to the Discussion, renumbering it as Section 5.1. (Line 530-552)

**Minor comments**

Line17: "limitations in modelling skills" in the abstract might be the defects or imperfections in the model algorithm, such as not involving lake processing. But the term "limitations in modelling skills" also covers the content of the forcing. So I suggest change "limitations in modelling skill" to "limitations in model capacity"

Reply: Thanks for the suggestion. We have revised it. The whole sentence is revised as follows.

"A major challenge lies in the limited quality of precipitation data and constraints in model capabilities." (Line 21-22)

Line34: Change "limitations of hydrological modelling" to "limited capacity of hydrological model".

Reply: Thanks and done. The sentence has been revised as follows.

"The credibility of hydrological simulations in data-scarce regions is challenged by the limited quality of precipitation data (e.g., incomplete, unreliability, and poor in-suit coverage) and the constrained capacity of the hydrological model given the underlay's complexities." (Line 40-42)

Line57: Change "realistic regional detail" to "the realistic regional detail"

Reply: Thanks and done. See Line 63.

Line58: Change "realistic regional details" to "refined-scale features"

Reply: Thanks and done. See Line 64.

Line67: Change "coarse resolution" to "the coarse resolution"

Reply: Thanks and done. (Line 72-74)

Line86: Change "suggesting" to "which suggests"

Reply: Thanks and done. Please see Line 92.

Line107-108: Please add references to the sentence "However, the region faces increasing risks of drought and flood, which are likely exacerbated by climate change."

Reply: Thanks for the comment. We have added the reference and revised the sentence as "However, the region is observed to be at risk of drought and flood, which are likely exacerbated by climate change (Kenya Climate Change Case Study, 2024)." (Line 111-113)

Line118-119: Change "The research is to improve hydrological models for better water resource management and risk mitigation, supporting sustainable practices in regions vulnerable to water-related damages." to "The research aims to improve hydrological models, which helps to better water resource management and risk mitigation, and

supports sustainable practices in regions vulnerable to water-related damages."

Reply: Thanks and done. Please see Line 123-124.

Line129: Please keep the hyphen consistent in "S 1.25°~N 0.50°, E 36.50°-E 39.75°".

Reply: Thanks for pointing out our carelessness. We have fixed it. Please see Line 133.

Line131-132: Before "We classified the terrain into mountainous regions above 1,600 meters and plains below 1,600 meters.", please add "To analyze and evaluate the spatial distribution of precipitation related to the topography,"

Reply: Thanks and we have revised. Very helpful suggestion. Please see Line 136-137.

Line133-135: Change "There are five reservoirs in the basin and along the Tana River (Table 1, Fig. 1 c). It is worth noting that the Garissa station is downstream Rukanga and the lakes between them are Masinga, Kamburu, Gitaru, Kindaruma, and Kiambere from the upstream to downstream. While the lakes don't affect the streamflow at Rukanga, they do impact the discharge at Garissa." to "There are five reservoirs in the basin and along the Tana River (Table 1, Fig. 1 c), including Masinga, Kamburu, Gitaru, Kindaruma, and Kiambere from the upstream to downstream. The five lakes are between Garissa station upstream and Rukanga downstream. It is worth noting that the lakes don't affect the streamflow at Rukanga while they do impact the discharge at Garissa."

Reply: Thanks for the suggestions. To make the context coherent, we have changed a bit and revised it as "There are five reservoirs in the basin along the Tana River, including Masinga, Kamburu, Gitaru, Kindaruma, and Kiambere from the upstream to downstream (Table 1, Fig. 1 c). These five lakes are between Garissa station upstream and Rukanga downstream. It is important to note that the lakes don't affect the streamflow at Rukanga, but they do impact the discharge at Garissa." Please see Line 138-140.

Line143: Change "boundaries" to "boundary"

Reply: Thanks and done. (Line 144)

Line154: If one year is used as the spin-up year, the subsequent analysis in the results

should be based on the simulated precipitation from 2011 to 2014 (which is 2010-2014 in the paper). However, I think the analysis in the results covering 2010 to 2014 is acceptable due to three reasons: 1) one month of spin-up is typically sufficient for WRF downscaling, 2) the time span of the precipitation data used is rather limited, 3) the precipitation analysis or evaluation focus on the MAM and OND, or the extremes. If you can extend the WRF downscaling (for example, from 2001 to 2014), it would be better.

Reply: Thanks for pointing it out. We delete "with the first year of spin-up." Sect. 3.1 has been changed to as follows.

"To obtain convection-permitting regional climate model simulations, we used the Advanced Research WRF (WRF-ARW) model of version 4.4 (Skamarock et al., 2019) with the designed domain of 5 km spatial resolution (Fig. 1). The lateral boundaries were forced with the 6-hourly ERA5 reanalysis with a spatial resolution of 0.25 degrees (Hersbach et al., 2020). The model was set with 50 vertical levels up to 10hPa. The convection parameterization was turned off for the CPWRF simulation, the Mellor-Yamada Nakanishi Niino Level 2.5 (MYNN2.5) Scheme (Nakanishi and Niino, 2006) for the planetary boundary layer, the RRTM scheme for longwave radiation (Mlawer et al., 1997), and the Dudhia Shortwave scheme for shortwave radiation (Jimy Dudhia, 1989). The Noah-MP Land Surface model ('Noah-MP LSM', Yang et al., 2011) was used for the land surface scheme.

The model runs from 1 January 2010 to 31 December 2014. Typically, WRF simulations require a spin-up of about one month, which should ideally be excluded from precipitation evaluation. However, given the limited length of simulated precipitation, the subsequent analysis is based on full precipitation simulation from January 2010 to December 2014." (Line 154-163)

Line367: Change "GIS pre-41.processing" to "GIS pre-processing"

Reply: Thanks for the careful review. We have revised the sentence. (Line 385-386)

Line376: Change " in the evolution of discharge " to " based on the discharge "

Reply: Thanks for the suggestion.

The sentence "The spin-up sensitivity is highlighted in the evolution of discharge during 2011-2014 from the 17 spin-up experiments (Fig. 5 and Table 3)." has been revised as follows.

"In the LakeRaw simulation, the spin-up sensitivity is highlighted by the discharge during 2011-2014 from the 17 spin-up experiments (Fig. 5 and Table 3)." (Line 394-395)

Line378: There are two things related to the term "peak-flow". One is the largest simulated daily flow on the date when the largest daily flow occurs based on the observations. Another is the peaks shown in the discharge-date curve. Please distinguish between the two.

Reply: Thanks for your helpful suggestion. We have changed the first term "peak-flow" to "Peak-Flow". The term "Peak-Flow" has been clearly defined in Set. 3.3 (Line 285-289). In the whole text, the term related to "peak-flow" has been changed to "Peak-Flow".

Line408: Change "demonstrate" to "demonstrates"

Reply: Thanks and done. Please see Line 426.

Line411: Please add references (i.e. figures) for "The water levels of the five lakes show the same spin-up time."

Reply: Thanks and done. Please see Line 429.

Line411-412: Change "However, a larger lake seems to require more time to reach equilibrium. The lakes are interconnected, so the initialization time is determined by the longest spin-up. Therefore, despite the disparate sizes, the initialization times of the five lakes are the same." to "Usually, a larger lake seems to require more time to reach equilibrium. Since the lakes are interconnected, the initialization time is determined by the longest spin-up. This results in the same initialization times of the five lakes despite of the dramatically disparate sizes".

Reply: Thanks for the suggestion. After combining this suggestion with the comments of the Reviewer 2, we revised the related paragraph as follows.

"The water levels from the lake/reservoir-integrated model show a consistent spin-up

period of 4 years across nearly all five lakes for the entire period, as well as during both the rainy and dry seasons (Fig. S2). Although KIAMBERE (one of the five lakes) exhibits a spin-up period of 3 years during the rainy season (Fig. S2 e), it can be considered nearly 4 years due to the uncertainty in determining the spin-up time required for the stabilization of specific variables. Since the lakes are interconnected, the stabilization time is governed by the longest spin-up period. This may result in nearly the same initialization time for all five lakes (Table 1)." Please see Line 428-433.

Line413-415: Please delete "The bias from the LakeRaw simulation is considerable (> 80 %). This is due to that the parameters used in the LakeRaw are from the primarily calibrated LakeNan Model or GIS pre-processing (Methodology), which needs further calibration for the WRF-Hydro system." Because there is no relationship between this part and the preceding text.

Reply: Thanks and done. Please see Line 488-493.

Line492: Please delete "still".

Reply: Thanks and done. Please see Line 507.

Line499: Please double-check the figure "-53".

Reply: Thanks and revised. Please see Line 514.

Line527: Change "under estimation" to "underestimation"

Reply: Thanks and revised. Please see Line 540.

Line532: The label in Fig. 10 is a bit confusing. Please label them more clearly.

Reply: Thanks for the useful suggestion and we have revised it. Please see Line 545-552.

Line551: Change "Hydrological modelling improvement from convection-permitting WRF-simulated precipitation" to "Hydrological modelling improvement from lake/reservoir module"

Reply: Thanks for your careful review and we have revised it. Please see Line 574.

Line554: Delete ". Factors".

Reply: Thanks and done. Please see Line 576.

**Response to Reviewer #2:**

**Overview of the paper**

The paper presents a hydrological simulation in a data-scarce region, utilizing a convection-permitting climate model and a lake-integrated hydrological model. It highlights and quantifies the improvements brought by convection-permitting WRF simulations and the inclusion of lake and reservoir modeling. A key contribution of the paper is its identification of specific areas of improvement, particularly at peak and low-flow points, and a detailed explanation of the underlying causes and attributions of these improvements.

The study addresses the added value of convection-permitting modeling in hydrological simulations and the integration of the lake/reservoir module within WRF-Hydro, both of which are topics that have been rarely explored. The findings provide fresh insights into the benefits of using a convection-permitting climate model and the lake/reservoir module, offering valuable benchmarks for optimizing hydrological modeling, especially in regions with limited data availability. As noted by the authors, this enhanced hydrological simulation has potential future applications in forecasting extreme hydrological events, such as floods and droughts.

The paper aligns well with the scope of HESS, as it focuses on hydrological simulation, especially in a data-scarce region. The use of the convection-permitting climate model and lake-integrated hydrological model addresses critical concerns in the climate-hydrology field. Furthermore, the study area, East Africa, is particularly relevant due to the scarcity of data, the complexity of hydrological simulations, and the frequent occurrence of extreme flood and drought events.

In conclusion, this paper offers significant value and is suitable for publication in HESS. However, I have several suggestions and comments for consideration, as outlined below.

**Major comments**

Although this manuscript is well-structured and comprehensive, with some well-written sections, it requires careful editing by professional English editors. Special attention

should be given to sentence structure, as well as minor spelling and grammatical errors, to ensure that the study's goals and results are clear to the reader.

Reply: Thanks for your comment. We have revised the English language throughout the manuscript to enhance clarity and readability.

The conclusions should be presented with caution. The sensitivity of the simulated peak flows to the spin-up time, based on a single event in 2011. The conclusion may vary if different regions or other peak flow events are considered. I recommend adding further discussion on this point.

Reply: Thank you for the suggestion. We added further discussion on this point in Sect. 5.4. The added discussion now reads as follows.

"Also, uncertainty may exist in the sensitivity analysis of the simulated peak flow to spin-up time, which was based on a single event (the largest observed peak from 2010 to 2014) at a specific discharge station (i.e., Garissa or Rukanga). The conclusion, especially about the spin-up time required for model stabilization, may vary when different regions or other peak flow events are considered. For example, a WRF/WRF-Hydro simulation (Lu et al., 2020) exhibits that initialization times needed for soil moisture stabilization differ for different basins in Western Norway. The varying spin-up periods required for flow stabilization between the dry and rainy seasons (Sect. 4.2.2 and Fig. 5 d-e) indicate the possible sensitivity of peak flow to spin-up duration across different peak flow events. The sensitivity of different regions and other peak flow events to spin-up time will be further investigated." (Line 619-626)

The authors should verify the model configuration. Figure 1 indicates that WRF is directly driven by ERA5, but lines 156-157 suggest a nested simulation domain. Please double-check this for accuracy.

Reply: Thanks for your careful review. We have corrected it in the revised manuscript as below.

"The convection parameterization was turned off for the WRF simulation,…". (Line 157)

When evaluating the WRF simulation, the authors focus primarily on bias. However, a

smaller bias does not necessarily indicate a better simulation. I suggest using additional indices, such as a Taylor diagram, to support the conclusions.

Reply: Thanks for your valuable suggestion. We acknowledge that focusing only on bias may not fully capture the simulation's accuracy. We have added additional evaluation indices, including the Taylor diagram (Fig. S1), to provide a more comprehensive assessment of the WRF simulation's performance. To ensure the continuity of the context, we have revised Section 4.1 (Line 315-367). In addition, we have updated the methodology in Sect. 3.4 to include details on the Taylor Diagram(Line 300-302), and added discussion about the performance of the model in the Uncertainty (Line 611-618). The revised texts can be seen as follows.

**"4.1.    WRF Precipitation refinement**

[revised manuscript text omitted]

The manuscript emphasizes the importance of CPM in East Africa. If the authors add a discussion of the added values of CPM with respect to its driving forces, this manuscript would be more informative.

Reply: Thank you for this insightful suggestion. To address this, we have expanded Section 5.2 including the added value of Convection-Permitting Models (CPM) in East Africa, in relation to their driving forces (ERA5). This additional content emphasizes that CPM enhances the representation of convective processes compared to coarser-resolution models, over Mount Kenya and the surrounding area. To ensure the continuity of the context, we have revised Section 5.2 as follows (Line 554-573) and added a description of the AV (added value) methodology in Sect. 3.4 (Line 292-302), which are as follows.

"**5.2.    Hydrological modeling improvement from CPWRF precipitation**

Dynamic downscaling at convection-permitting resolution allows for a more accurate representation of precipitation processes. The CPWRF simulation enhances local (e.g., mesoscale) processes and interactions between local and large-scales, especially over complex terrain (Kendon et al., 2021; Guevara Luna et al., 2020; Schmidli et al., 2006; Schumacher et al., 2020; Li et al., 2020). As a result, CPWRF potentially contributes to improving precipitation simulation in our study (Sect. 4.1), especially reducing bias in seasonal precipitation over mountainous areas, and light rainfall (1-15mm day-1) probability in the dry season compared to ERA5 (Fig. 3 and Table 8).

The improvement in the seasonal precipitation over mountainous regions and rainfall probability can be supported by the spatial distribution of the added value (AV) in seasonal precipitation with respect to the driving forces (Fig. S2). The CPWRF simulation adds consistent value to ERA5 over the mountainous areas across all four seasons (MAM, OND, JF, and JJAS). The area with positive AV is mainly over Mount Kenya and its surrounding areas, with the positive AV being particularly distinct during

the dry season. CPWRF also adds value to ERA5 in the light rainfall probability (Fig. S2 f-k), as demonstrated in Sect. 4.1. The basin averaged AV of CPWRF over the probability of light precipitation events are 0.32, 0.26, 0.30, and 0.07 in MAM, OND, JF, and JJAS, respectively. The positive AV of CPWRF with respect to ERA5 over the extreme rainfall probability, also concentrates around Mount Kenya, consistently across all four seasons (Fig. S2 l-p). Previous studies (Giorgi et al., 2022) have demonstrated that the added value of CPWRF simulations is influenced by various factors, including timescale, variables, regions, and uncertainty of the benchmark. Therefore, further in-depth research is required for a more reliable AV assessment.

Due to the precipitation improvement from WRF, hydrological simulation with CPWRF precipitation as a driving force (LakeCal), showed significant improvements, compared to simulations driven by ERA5 (LakeCal-ERA5) (Fig. 10a). These improvements are particularly notable in reducing false peak simulations, likely due to the reduction in the overestimation of light rainfall probability. The enhancement in peak flow simulation is also observed in the WRF-Hydro model without the lake/reservoir module (Fig. 10b)." (Line 554-573)

**3.4. Evaluation of simulated precipitation from CPWRF**

To assess whether the CPWAR has advantages over their driving forces (ERA5), added value (AV) proposed by Dosio et al. (2015) was applied, expressed as follows.

$$AV = \frac{(X_{ERA5} - X_{IMERG})^2 - (X_{CPWRF} - X_{IMERG})^2}{\max\left((X_{ERA5} - X_{IMERG})^2, (X_{CPWRF} - X_{IMERG})^2\right)}$$

(3)

*$X_{ERA5}$, $X_{CPWRF}$, and $X_{IMERG}$* indicate precipitation from the driving forces (ERA5), CPWRF simulation, and benchmark (IMERG), respectively. The added value (AV) from CPWRF is defined as the performance difference between itself and the driving forces for precipitation in a specific region and period. If the CPWRF adds value to the driving forces (ERA5), the AV is positive, whereas a negative AV suggests no adding value.

To fully evaluate the simulated precipitation by CPWRF, we also employed Taylor diagrams (Taylor, 2001), which present a concise statistical summary in terms of spatial correlation (indicated by correlation coefficient), and spatial variance (indicated by normalized standardized deviation). A higher spatial correlation and a spatial variance closer to one indicate better simulation skills." (Line 292-302)

**Minor comments**

Line17-18: Replace "the upper and middle stream of the Tana River basin was" with "the upper and middle streams of the Tana River basin were ".

Reply: Thanks for the suggestions. We have revised it as "the upper and middle Tana River basin". (Line 22-23)

Line19-20: This sentence is ambiguous. Please revise it.

Reply: Thank you for pointing it out. We have revised the sentence and changed it to "We performed convection-permitting (CP) simulations using the Weather Research and Forecasting (WRF) model, and utilizing the CPWRF output as a driver we conducted WRF Hydrological modelling (WRF-Hydro) integrated with the lake/reservoir module." (Line 20-22)

Line21: Replace "using IMERG as the benchmark" with "when benchmarked against IMERG

Reply: Thanks and done. Please see Line 26-27.

Line22-23 Change "alleviates the peak false" to "alleviates the false peak simulation".

Reply: Thanks for the suggestion. We have revised it as "reducing false peak occurrences". Please see Line 28-29)

Line24: Change "NSE" with "NSE (Nash-Sutcliffe Efficiency)".

Reply: Thanks for the suggestion. We have revised it as "Nash-Sutcliffe Efficiency (NSE)". Please see Line 29.

Lin29-30: There are two terms "lake" and "lake/reservoir" which seem to represent the same thing. It would be better to unify them for consistency throughout the document.

Reply: Thanks and done.

Line29-30: Replace "highlight the enhanced hydrological modelling capability with" with "highlight the enhanced capability of hydrological modelling using".

Reply: Thank you for the suggestion. We have revised the sentence "Our findings highlight the enhanced hydrological modelling capability with the lake/reservoir module and CPWRF simulations, offering a valuable tool for flood and drought predictability in data-scarce regions such as East Africa." as follows.

"These findings highlight the enhanced capability of hydrological modeling when combining CPWRF simulations with lake/reservoir module, providing a valuable tool for improving flood and drought predictability in data-scarce regions like East Africa." (Line 36-38)

Line123: Replace "resulting" with "which results".

Reply: Thanks and done. Please see Line 128.

Line129: Please replace "S 1.25°~N 0.50°" with "S 1.25°-N 0.50°".

Reply: Thanks and done. Please see Line 133.

Line139: Replace " the upper and middle stream of the Tana River Basi" with " the upper and middle streams of the Tana River basin".

Reply: Thanks for the suggestion. We have revised it as "the upper and middle Tana River Basin" (Line 144).

Line:154: Usually, one month spin-up is sufficient for WRF downscaling.

Reply: Thanks for pointing it out. We have revised Sect. 3.1 as the follows.

"To obtain convection-permitting regional climate model simulations, we used the Advanced Research WRF (WRF-ARW) model of version 4.4 (Skamarock et al., 2019) with the designed domain of 5 km spatial resolution (Fig. 1). The lateral boundaries were forced with the 6-hourly ERA5 reanalysis with a spatial resolution of 0.25 degrees (Hersbach et al., 2020). The model was set with 50 vertical levels up to 10hPa. The convection parameterization was turned off for the CPWRF simulation, the Mellor-Yamada Nakanishi Niino Level 2.5 (MYNN2.5) Scheme (Nakanishi and Niino, 2006) for the planetary boundary layer, the RRTM scheme for longwave radiation (Mlawer et al., 1997), and the Dudhia Shortwave scheme for shortwave radiation (Jimy Dudhia,

1989). The Noah-MP Land Surface model ('Noah-MP LSM', Yang et al., 2011) was used for the land surface scheme.

The model runs from 1 January 2010 to 31 December 2014. Typically, WRF simulations require a spin-up of about one month, which should ideally be excluded from precipitation evaluation. However, given the limited length of simulated precipitation, the subsequent analysis is based on full precipitation simulation from January 2010 to December 2014." (Line 153-163)

Line182-183: Please add "(with the lake/reservoir module inactive)" behind "without the lake/reservoir module".

Reply: Thank you for the suggestion. We have revised it as requested. (Line 183-184)

Line185: Replace "of the Garissa discharge." with "of simulated discharge against the observation at Garissa".

Reply: Thank you for the suggestion. We have revised it. Please see Line 186-187.

Line208-210: Replace "which may affect the subsequent sensitivity analysis and hydrological modelling assessments." with "which may potentially affect the result of subsequent sensitivity analyses and the performance of the hydrological simulation."

Reply: Thank you for the suggestion. Since we have added new content here, to enhance the overall coherence, we have revised it as " which potentially influences the fidelity of model simulations (Ajami et al., 2014a; Ajami et al., 2014b; Bonekamp et al., 2018; Seck et al., 2015), and subsequently affect the result of subsequent sensitivity analyses and the performance of the hydrological simulation." (Line 213-215)

Line260: Replace "For each lake test" with "For each test".

Reply: Thanks for the suggestion. We have revised the sentence "For each lake test, there is a set of more than 30 simulations." to "For each test of parameters related to one lake, more than 30 simulations were conducted." (Line 263-264)

Line274: Replace "Each lake" with "Each".

Reply: Thanks for the suggestion. To make it clearer and improve the coherence of the context, we have revised the related sentence to "Of the lake/reservoir parameter sets, each was calibrated sequentially from upstream to downstream, with more than 30

experimental iterations." (Line 277-278)

Line306-307: Please change the unit "mm a$^{-1}$" to "mm". The unit should be corrected in the whole text.

Reply: Thanks and done. Please see Line 335.

Line368: Replace "GIS pre-41.processing" with "GIS pre-processing".

Reply: Thanks for the careful review. We have deleted the phrase.

The sentence "The hydrological parameters, which were based on the model without lake/reservoir module (LakeNan), and the groundwater component and lake-related parameter set as the default from GIS pre-41.processing (Methodology), need to be re-tuned when the lake/reservoir is included in WRF-Hydro system." has been revised as follows.

"The hydrological parameters, groundwater component and lake/reservoir-related parameters need to be further adjusted when the lake/reservoir is included in WRF-Hydro system." (Line 385-386)

Line408: Replace "demonstrate" with "demonstrates".

Reply: Thanks and done. Please see Line 426.

Line411: The view "a larger lake seems to require more time to reach equilibrium." depends. You should add some references.

Reply: Thanks for the helpful suggestions. We did not find a related reference. So, we have removed this sentence and revised the related paragraph as follows.

"The water levels from the lake/reservoir-integrated model show a consistent spin-up period of 4 years across nearly all five lakes for the entire period, as well as during both the rainy and dry seasons (Fig. S2). Although KIAMBERE (one of the five lakes) exhibits a spin-up period of 3 years during the rainy season (Fig. S2 e), it can be considered nearly 4 years due to the uncertainty in determining the spin-up time required for the stabilization of specific variables. Since the lakes are interconnected, the stabilization time is governed by the longest spin-up period. This may result in nearly the same initialization time for all five lakes (Table 1)." (Line 428-433)

Line499: Replace "-53" with "-53%".

Reply: Thanks for pointing out our careless, and done. (Line 514)

Line527: Replace "under estimation" with "underestimation".

Reply: Thanks and done. (Line 540)

Line529: "WRF-refined precipitation" is not idiomatic. Please revise it.

Reply: Thanks for the suggestion. We have revised it as "the precipitation simulated by CPWRF". (Line 543)

Line544: Replace "Woodhams et al.'s research (2018) demonstrates" with "Woodhams et al. (2018) demonstrates".

Reply: Thanks for the suggestion. We have deleted this sentence (Line 554-573). Woodhams et al. (2018) describe the advantage of the convection-permitting WRF model on sub-daily time scale. It is not related to our study which focuses on a daily scale.

Line554: Delete ". Factors".

Reply: Thanks and done. Please see Line 576.

Line565: Replace "Arnault et al.," with "Arnault et al."

Reply: Thanks and done. (Line 586)

Line574-575: Replace "lake water levels may be not well presented" with "it shows a limited skill for simulating water level".

Reply: Thanks for the suggestion. We have revised the sentence. To improve the overall logic of the text, we have relocated the sentence to Line 630.

Line577: Please replace " with $r^2$ ranging from near zero (0.005) to 0.25 for the five lakes." with " with $r^2$ of the simulated discharge against the observation at Garissa less than 0.25 for all the five lakes."

Reply: Thanks for the suggestion. We have deleted the texts in the revised manuscript.

Line599: Replace "a seamless, consistent meteorological-hydrological modelling system" with "a seamless and consistent meteorological-hydrological modelling system".

Reply: Thanks and done. (Line 640)

Line601-602: The sentence "which makes an NSE increase of 0.53 when comparing

LakeCal to LakeCal-ERA5" is not clear. Please correct it.

Reply: Thanks for pointing it out.

We have revised the text "The refined precipitation from CPWRF simulation improves the hydrological simulation, which makes an NSE increase of 0.53 when comparing LakeCal to LakeCal-ERA5, contributing to a 24 % enhancement in the hydrological simulation." as follows.

"Compared to ERA5-driven simulation (LakeCal-ERA5), the CPWRF-driven WRF-Hydro simulation (LakeCal) increases NSE by 0.53, contributing to a 24 % improvement in the hydrological simulation." (Line 643-644)

We sincerely appreciate your valuable feedback and careful review of our manuscript. Your suggestions have helped us improve the clarity and depth of our work. Thank you for your time and effort in reviewing our paper!